# *Fantastic Weights and How to Find Them:*
# Where to Prune in Dynamic Sparse Training

**Aleksandra I. Nowak***
Jagiellonian University, Doctoral School of Exact and Natural Sciences;
IDEAS NCBR
`aleksandrairena.nowak@docotral.uj.edu.pl`

**Bram Grooten**
Eindhoven University of Technology

**Decebal Constantin Mocanu**
University of Luxembourg;
Eindhoven University of Technology;
University of Twente

**Jacek Tabor**
Jagiellonian University,
Faculty of Mathematics and Computer Science

## Abstract

Dynamic Sparse Training (DST) is a rapidly evolving area of research that seeks to optimize the sparse initialization of a neural network by adapting its topology during training. It has been shown that under specific conditions, DST is able to outperform dense models. The key components of this framework are the pruning and growing criteria, which are repeatedly applied during the training process to adjust the network's sparse connectivity. While the growing criterion's impact on DST performance is relatively well studied, the influence of the pruning criterion remains overlooked. To address this issue, we design and perform an extensive empirical analysis of various pruning criteria to better understand their impact on the dynamics of DST solutions. Surprisingly, we find that most of the studied methods yield similar results. The differences become more significant in the low-density regime, where the best performance is predominantly given by the simplest technique: magnitude-based pruning.

## 1 Introduction

Modern deep learning solutions have demonstrated exceptional results in many different disciplines of science [5, 14, 26]. However, they come at the cost of using an enormous number of parameters. Consequently, compression methods aim to significantly reduce the model size without introducing any loss in performance [20, 6].

One approach to sparsifying neural networks is pruning, in which a portion of the weights is removed at the end of the training based on some predefined importance criterion [31, 21, 20]. More recently, [15] have found that iterative pruning joined with cautious parameter re-initialization can identify sparse subnetworks that are able to achieve similar performance to their dense counterparts when trained from scratch. This result, known as the *lottery ticket hypothesis*, has launched subsequent research into methods for identifying and training models that are sparse already at initialization [32, 60, 66, 48, 54].

---

*Work started while visiting University of Twente.

37th Conference on Neural Information Processing Systems (NeurIPS 2023).

An especially promising direction in obtaining well-performing sparse neural networks is the Dynamic Sparse Training (DST) framework. Inspired by the neuroregeneration in the brain, DST allows for plasticity of the initial sparse connectivity of the network by iteratively pruning and re-growing a portion of the parameters in the model [42]. This relatively new concept has already gained increasing interest in the past few years. Most notably, in computer vision, DST demonstrated that it is sufficient to use only $20\%$ of the original parameters of ResNet50 to train ImageNet without any drop in performance [12, 38]. Even more intriguingly, in Reinforcement Learning applications, DST is able to significantly outperform the classical dense models [18, 58]. At the same time, general sparse networks have been reported to surpass their dense counterparts in terms of adversarial robustness [47]. All those results demonstrate the incredible potential of DST not only in increasing the model's efficiency but also in providing a better understanding of the features and limitations of neural network training.

Motivated by the above, we take a closer look at the current DST techniques. Most of the research concerning design choices in DST focuses on the analysis of methods where variations occur in the growth criterion, while the pruning criterion becomes intertwined with other design considerations [3, 12, 11, 1]. In this study, we address this issue by taking a complementary approach and focusing on the pruning criteria instead. A pruning criterion in DST serves as a measure of weight importance and hence is a proxy of the "usefulness" of a particular connection. Note that the importance of a connection in DST can differ from standard post-training pruning, as the role of weights can change throughout training due to the network's plasticity and adaptation. Weights deemed unimportant in one step can become influential in later phases of the training.

Our goal is to provide a better understanding of the relationship between pruning criteria and the dynamics of DST solutions. To this end, we perform a large empirical study including several popular pruning criteria and analyze their impact on the DST framework on diverse models. We find that:

- Surprisingly, within a stable DST hyperparameter setup, the majority of the studied criteria perform similarly, regardless of the model architecture and the selected growth criterion.
- The difference in performance becomes more significant in a very sparse regime, with the simplest magnitude-based pruning methods surpassing any more elaborate choices.
- Applying only a few connectivity updates is already enough to achieve good results. At the same time, the reported outcomes surpass those obtained by static sparse training.
- By analyzing the structural similarity of the pruned sets by each criterion, we assert that the best-performing methods make similar decision choices.

The insights from our research identify that the simplest magnitude pruning is still the optimal choice, despite a large amount of alternatives present in the literature. This drives the community's attention to carefully examine any new adaptation criteria for Dynamic Sparse Training.[2]

## 2   Related Work

**Pruning and Sparse Training.** A common way of reducing the neural network size is pruning, which removes parameters or entire blocks of layers from the network. In its classical form, pruning has been extensively studied in the context of compressing post-training models [25, 44, 31, 21, 20, 46, 33, 43, 17] – see e.g. [23, 4] for an overview and survey. Interestingly, [40] demonstrated for the first time in the literature that a sparse neural network can match and even outperform its corresponding dense neural network equivalent if its sparse connectivity is designed in a sensitive manner. Recently, a similar result was obtained by the *lottery ticket hypothesis* and follow-up research, which showed that there exist sparse subnetworks that can be trained in isolation to the same performance as the dense networks [15, 66, 62]. In an effort to find these subnetworks without the need for dense training, different techniques have been proposed over the years [32, 60, 59], including random selection [48, 34]. Such approaches are commonly referred to as Sparse Training, or Static Sparse Training to emphasize that the sparse structure stays the same throughout the training. Interestingly, [16] find out that the sparse initializations produced by the static approaches are invariant to parameter reshuffling and reinitialization within a layer and that even pre-training magnitude pruning performs quite well in such a setup.

---

[2]The code is provided at `https://github.com/alooow/fantastic_weights_paper`.

Table 1: An overview of the existing **pruning** criteria in DST. Our work analyzes the differences and similarities between all the methods and fills the gaps ($\times$) in the literature. Each column lists a pruning criterion, while each row presents a growing method. SNIP [32] was not designed to grow weights, but we investigate whether it can be applied in DST.

| | $\mathcal{C}_{\text{Magnitude}}$ $\|\theta\|$ | $\mathcal{C}_{\text{SET}}$ $\|\theta_+\|, \|\theta_-\|$ | $\mathcal{C}_{\text{MEST}}$ $\|\theta\| + \lambda\|\nabla_\theta \mathcal{L}(\mathcal{D})\|$ | $\mathcal{C}_{\text{Sensitivity}}$ $\frac{\|\nabla_\theta \mathcal{L}(\mathcal{D})\|}{\|\theta\|}$ | $\mathcal{C}_{\text{SNIP}}$ $\|\theta\|\|\nabla_\theta \mathcal{L}(\mathcal{D})\|$ |
|---|---|---|---|---|---|
| random growth | $\times$ | SET[42] | MEST[64] | Sensitivity[45] | $\times$ |
| gradient growth | RigL[12] | $\times$ | $\times$ | $\times$ | $\times$ |

**Dynamic Sparse Training.** In DST, the neural network structure is constantly evolving by pruning and growing back weights during training [42, 3, 11, 12, 64, 61, 1, 9, 28]. The key motivation behind DST is not only to provide compression but also to increase the effectiveness and robustness of the deep learning models without the need for overparametrization [38]. The capability of DST has recently been an area of interest. In [13], authors indicate that DST is able to outperform static sparse training by assuring better gradient flow. DST has also been reported to achieve high performance in Reinforcement Learning [18, 53, 19] and Continual Learning [52], with ongoing research into applicability in NLP [35]. Some attention has also been given to the topological properties of the connectivity patterns produced by DST. In particular, [36], investigated the structural features of DST. However, they focus only on one method, the Sparse Evolutionary Training (SET) procedure [42].

To the best of our knowledge, this work is the first to comparatively study a large number of pruning criteria in DST on multiple types of models and datasets. We hope that our analysis will increase the understanding within the dynamic sparse training community.

## 3 Background

### 3.1 Dynamic Sparse Training

Dynamic sparse training is a framework that allows training neural networks that are sparse already at initialization. The fundamental idea behind DST is that the sparse connectivity is not fixed. Instead, it is repeatedly updated throughout training. More precisely, let $\theta$ denote all the parameters of a network $f_\theta$, which is trained to minimize a loss $L(\theta; \mathcal{D})$ on some dataset $\mathcal{D}$. The density $d^l$ of a layer $l$ with latent dimension $n^l$ is defined as $d^l = \|\theta^l\|_0/n^l$, where $\|\cdot\|_0$ is the L0-norm, which counts the number of non-zero entries. Consecutively, the overall density $D$ of the model is $D = \frac{\sum_{l=1}^{L} d^l n^l}{\sum_{l=1}^{L} n^l}$, with $L$ being the number of layers in the network. The *sparsity* $S$ is given by $S = 1 - D$. Before the start of training, a fraction of the model's parameters is initialized to be zero in order to match a predefined density $D$. One of the most common choices for initialization schemes is the Erdős-Rényi (ER) method [42], and its convolutional variant, the ER-Kernel (ERK) [12]. It randomly generates the sparse masks so that the density in each layer $d^l$ scales as $\frac{n^{l-1}+n^l}{n^{l-1}n^l}$ for a fully-connected layer and as $\frac{n^{l-1}+n^l+w^l+h^l}{n^{l-1}n^l w^l h^l}$, for convolution with kernel of width $w^l$ and height $h^l$. The sparse connectivity is updated every $\Delta t$ training iterations. This is done by removing a fraction $\rho_t$ of the active weights accordingly to a pruning criterion $\mathcal{C}$. The selected weights become inactive, which means that they do not participate in the model's computations. Next, a subset of weights to regrow is chosen accordingly to a growth criterion from the set of inactive weights, such that the overall density of the network is maintained. The pruning fraction $\rho_t$ is often decayed by cosine annealing $\rho_t = \frac{1}{2}\rho(1 + \cos(t\pi/T))$, where $T$ is the iteration at which to stop the updates, and $\rho$ is the initial pruning fraction.

The used pruning and growth criterion depends on the DST algorithm. The two most common growth modes in the literature are random growth and gradient growth. In the first one, the new connections are sampled from a given distribution [42, 45, 64]. The second category selects weights based on the largest gradient magnitude [12]. Other choices based on momentum can also benefit the learning [11].

### 3.2 Pruning Criteria

Given a pruning fraction $\rho_t$, the pruning criterion $\mathcal{C}$ determines the importance score $s(\theta_{i,j}^l)$ for each weight and prunes the ones with the lowest score. We use *local pruning*, which means that the

criterion is applied to each layer separately.[3]. For brevity, we will slightly abuse the notation and write $\theta$ instead of $\theta_{i,j}^l$ from now on. Below, we discuss the most commonly used pruning criteria in DST that are the subject of the conducted study.

**SET.** To the best of our knowledge, SET is the first pruning criterion used within the DST framework, which was introduced in the pioneering work of [42]. The SET criterion prunes an equal amount of positive and negative weights with the smallest absolute value.

**Magnitude.** The importance score in this criterion is given by the absolute value of the weight $s(\theta) = |\theta|$. Contrary to SET, no division between the positive and negative weights is introduced. Due to its simplicity and effectiveness, the magnitude criterion has been a common choice in standard post-training pruning, as well as in sparse training [15, 12].

**MEST.** The standard magnitude criterion has been criticized as not taking into consideration the fluctuations of the weights during the training. The MEST (Memory-Economic Sparse Training) criterion [64], proposed to use the gradient as an indicator of the trend of the weight's magnitude, leading to a score function defined as $s(\theta) = |\theta| + \lambda|\nabla_\theta \mathcal{L}(\mathcal{D})|$, where $\lambda$ is a hyperparameter.

**Sensitivity.** The role of gradient information in devising the pruning criterion has also been studied by [45]. Taking inspiration from control systems, the authors propose to investigate the relative gradient magnitude in comparison to the absolute value of the weight, yielding $s(\theta) = |\nabla_\theta \mathcal{L}(\mathcal{D})|/|\theta|$. In our study, we consider the *reciprocal* version of that relationship $s(\theta) = |\theta|/|\nabla_\theta \mathcal{L}(\mathcal{D})|$, which we call *RSensitivity*, as we found it to be more stable.[4]

**SNIP.** The SNIP (Single-shot Network Pruning) criterion is based on a first-order Taylor approximation of the difference in the loss before and after the weight removal. The score is given by $s(\theta) = |\theta| \cdot |\nabla_\theta \mathcal{L}(\mathcal{D})|$. The criterion has been originally successfully used in static sparse training [32] and post-training pruning [43]. Motivated by those results, we are interested in investigating its performance in the dynamic sparse training scenario.

We denote the above-mentioned criteria by adding a subscript under the sign $\mathcal{C}$ in order to distinguish them from the algorithms that introduced them. We summarize their usage with random and gradient growth criteria in the DST literature in Table 1.

## 4 Methodology

This work aims to understand the key differences and similarities between existing pruning techniques in DST by answering the following research questions:

**Q1** What is the impact of various pruning criteria on the performance of dynamic sparse training?
**Q2** How does the frequency of topology updates influence the effectiveness of different pruning methods?
**Q3** To what extent do different pruning criteria result in similar neural network topologies?

**Q1: Impact on the Performance of DST.** In the first research question, we are interested in investigating the relationship between the pruning criterion and the final performance of the network. We compare the results obtained by different pruning criteria on a diverse set of architectures and datasets, ranging from a regime with a small number of parameters ($< 100K$) to large networks ($\sim 13M$). We examine multi-layer perceptrons (MLPs) and convolutional neural networks (ConvNets). Within each model, we fix the training setup and vary only the pruning criterion used by the DST framework. This allows us to assess the benefits of changing the weight importance criterion in isolation from other design choices. As the growing criterion, we choose the uniform random [42] and gradient [12] approaches, as they are widely used in the literature. In addition, we also perform a separate comparison on the ResNet-50 model ($25M$ parameters) on ImageNet.

The common sense expectation would be that the more elaborate pruning criteria, which combine the weight magnitude with the gradient information, will perform better than the techniques that rely only on the weight's absolute value. Since the gradient may provide information on how the weight will change in the future, it indicates the trend the weight might follow in the next optimization updates.

---

[3]We adapt this setup as it is commonly used in DST approaches, and has a reduced risk of entirely disconnecting the network – see Appendix I for a more detailed discussion.

[4]See Appendix B.

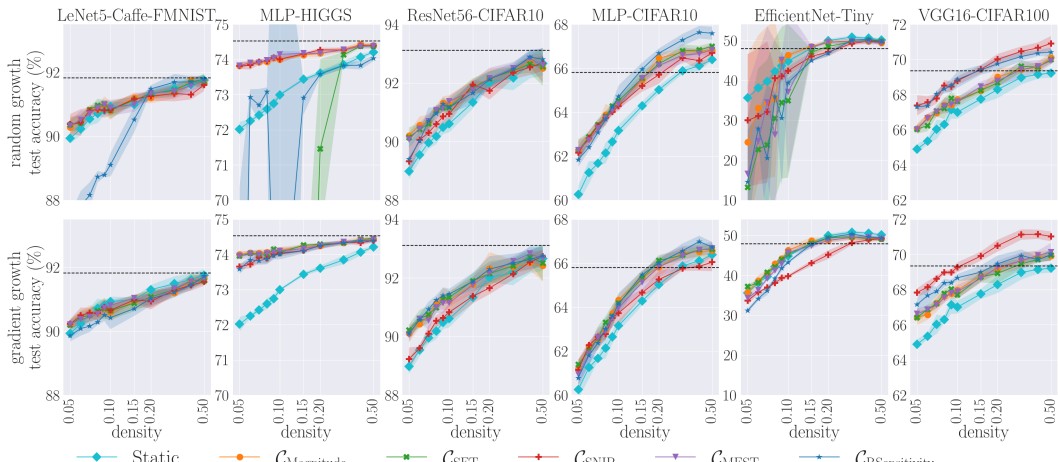

Figure 1: The test accuracy versus density computed on the studied models for different pruning criteria (due to space limits we only present here six plots, see Figure 10 in Appendix D for the remaining two setups). The first row represents the results obtained by random growth, while the second corresponds to gradient growth. Note the logarithmic scale in the x-axis. The performance of the dense model is indicated by the horizontal dashed line. In almost every case all pruning criteria perform well, regardless of the chosen model and growth criterion. At the same time, they surpass the static initialization (in light blue).

Therefore it may be more suitable in the DST approach, where connectivity is constantly evolving, making insights into future updates potentially beneficial. This is especially promising considering the effectiveness of the gradient-based approach in estimating the importance in the growth criterion.

**Q2: Update Period Sensitivity.** In the second research question, we focus on one of the most important design choices: the topology update period $\Delta t$. The setting of this update period determines how often the network structure is adapted during training. When considered together with the dataset–and batch-size, this hyperparameter can be interpreted as the number of data samples the model trains on before performing the mask update. Therefore, within a fixed dataset and batch size, using a relatively low value means that the topology is changed very frequently in comparison to the optimization update of the weights. This poses a potential risk of not letting the newly added connections reach their full potential. On the other hand, a high value gives the weights enough opportunity to become important but may not allow the sparse network to adjust its structure based on the data characteristics. Finding a balance between this exploitation-exploration conflict is an imperative task in any DST framework.

**Q3: Structural Similarity.** Finally, we take a deep dive into the structures produced by the various pruning criteria. Please note that while the previous questions, Q1 and Q2, may explain how the pruning criterion choice affects performance and update frequency in DST, they do not indicate whether the mask solutions obtained by those criteria are significantly different. In order to assess the diversity of the studied methods in terms of the produced sparse connectivity structures, we need to compare the sets of weights selected for pruning by each criterion under the same network state. In addition, we also investigate the similarity between the final masks obtained at the end of training and how close they are to their corresponding sparse initializations. This allows us to assess whether the DST topology updates are indeed meaningful in changing the mask structure. For both cases, we incorporate a common score of the proximity of sets, known as the Jaccard index (or Intersection-Over-Union). We compute it separately for each layer and average the result:

$$\bar{J}(I_a, I_b) = \frac{1}{L} \sum_{l=1}^{L} J(I_a^l, I_b^l), \quad J(I_a^l, I_b^l) = \frac{|I_a^l \cap I_b^l|}{|I_a^l \cup I_b^l|}, \tag{1}$$

where $I_a^l$ and $I_b^l$ are the sets selected by pruning criteria $a$ and $b$ in layer $l$. A Jaccard index of $1$ indicates that sets overlap perfectly, while $0$ implies they are entirely separate.

We expect to see some overlap in the pruning methods' selection of weights since all of them incorporate the use of the magnitude of the weight. We also hope to see if the difference between the connectivities produced by scores that give different performances is indeed large or whether a small adjustment of the weights would render them equal.

## 5   Experiments

In this section, we present the results of our empirical studies. We start with the description of the experimental setup and then answer each of the posed questions in consecutive sections.

### 5.1   Setup of the Experiments

We perform our analysis using eight different models, including small- and large-scale MLPs and Convolutional Nets. The small-MLP is a 4-layer network with hidden size of at most $256$, (trained on the tabular Higgs dataset [2]). The large-MLP also consists of 4 layers (latent dimension size up to $1024$ neurons) and is evaluated on the CIFAR10 [27]. The convolutional architectures are: a small 3-layer CNN (CIFAR10), LeNet5-Caffe [30] (FashionMNIST [63]), ResNet56 [22](CIFAR10, CIFAR100), VGG-16 [50](CIFAR100) and EfficientNet [57](Tiny-ImageNet [29]). In addition, on a selected density, we also consider the ResNet50 model on ImageNet [49]. We summarize all the architectures in Appendix A. While our primary focus is on vision and tabular data, we also evaluate the pruning criteria on the fine-tuning task of ROBERTa Large [39] using the CommonsenseQA dataset[56] adapted from the Sparsity May Cry (SMC) Benchmark [35]. In contrast to the setup studied throughout this work, this experiment involves a fine-tuning type of problem. In consequence, we discuss it in Appendix C and focus on training models from scratch for the remainder of the paper.

We train the models using the DST framework on a set of predefined model densities. We use adjustment step $\Delta t = 800$ and initial pruning fraction $\rho = 0.5$ with cosine decay. As the growth criterion, we investigate both the random and gradient-based approaches, as those are the most common choices in the literature. Following [64], we reuse the gradient computed during the backward pass when calculating the scores in the gradient-based pruning criteria. In all cases, we ensure that within each model, the training hyperparameters setup for every criterion is the same. Each performance is computed by averaging 5 runs, except for ImageNet, for which we use 3 runs. The ResNet50 model is trained using density 0.2 and gradient growth (we select the gradient-based growth method since is known to provide good performance in this task [12]). Altogether we performed approximately **7000** runs of dense, static sparse, and DST models, which in total use 8 (9 including ImageNet) different dataset-model configurations, and jointly took approximately **5275** hours on GPU.[5]

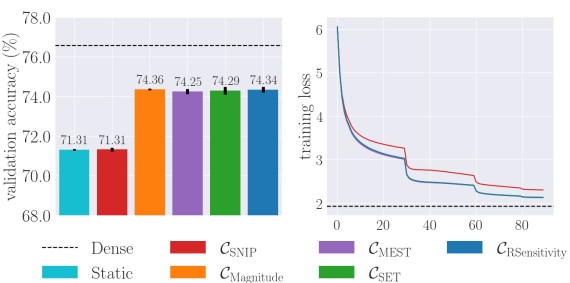

Figure 2: **Left:** Validation accuracy of the different pruning criteria on ImageNet obtained for density $0.2$. The dashed black line indicates the best result of the dense model. **Right:** The training loss versus the number of epochs. We see that all methods perform similarly, except for $\mathcal{C}_{\text{SNIP}}$, which suffers already at the beginning of the training.

### 5.2   Impact on the Performance of DST

In this section, we analyze the impact of the different pruning criteria on the test accuracy achieved by the DST framework. We compare the results with the original dense model and static sparse training with ERK initialization. The results are presented in Figure 1.

In this work, we are interested in how the frequency of the topology update impacts different pruning criteria. Intuitively, one could anticipate that the pruning criteria incorporating the gradient

---

[5]For training details please refer to Appendix A.2.

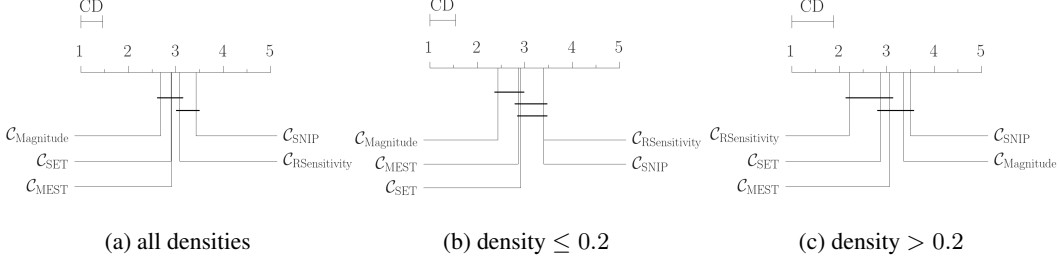

(a) all densities       (b) density $\leq 0.2$       (c) density $> 0.2$

Figure 3: The critical distance diagram of the studied methods for **(a)** all densities, **(b)** densities smaller or equal $0.2$, and **(c)** densities larger than $0.2$. For each model we compute its average rank (the lower the better) and plot it on the horizontal line. The methods with ranks not larger than the critical distance (denoted as CD, displayed in the left upper corner of each plot) are considered not statistically different and are joined by thick horizontal black lines.

information will perform better with a smaller update period, as the gradient may indicate the trend in the weight's future value.

Firstly, we observe that the differences between the pruning criteria are usually most distinctive in low densities. Furthermore, in that setting, the dynamic sparse training framework almost always outperforms the static sparse training approach. This is important, as in DST research, high sparsity regime is of key interest. Surprisingly, the more elaborate criteria using gradient-based approaches usually either perform worse than the simple magnitude score, or do not hold a clear advantage over it. For instance, the Taylor-based $\mathcal{C}_{\text{SNIP}}$ criterion, despite being still better than the static initialization, consistently achieves results similar or worse than $\mathcal{C}_{\text{Magnitude}}$. This is especially visible in the DST experiments using gradient growth (note ResNet56, MLP-CIFAR10, and EfficientNet). The only exception is the VGG-16 network. In addition, we also observe that $\mathcal{C}_{\text{SNIP}}$ gives better performance when used in a fine-tuning setup for EfficientNet - see Appendix H. Consequently, this Taylor-approximation-based criterion, although being well suited for standard post-training pruning [43] and selecting sparse initialization masks [32] seems not to be the optimal choice in dynamic sparse training approaches on high sparsity. The $\mathcal{C}_{\text{RSensitivity}}$, on the other hand, can outperform other methods but is better suited for high densities. Indeed, it is even the best choice for densities larger than $0.15$ for large-MLP on CIFAR10 and ResNet models. Finally, the $\mathcal{C}_{\text{MEST}}$ criterion also is not superior to the $\mathcal{C}_{\text{Magnitude}}$, usually leading to similar results.[6] We also present the results obtained for ResNet-50 on ImageNet with density $0.2$ in Figure 2. We observe that, again, the $\mathcal{C}_{\text{SNIP}}$ criterion leads to the worst performance. The $\mathcal{C}_{\text{RSensitivity}}$, $\mathcal{C}_{\text{SET}}$, and $\mathcal{C}_{\text{MEST}}$ methods achieve high results but not clearly better than the $\mathcal{C}_{\text{Magnitude}}$ criterion.

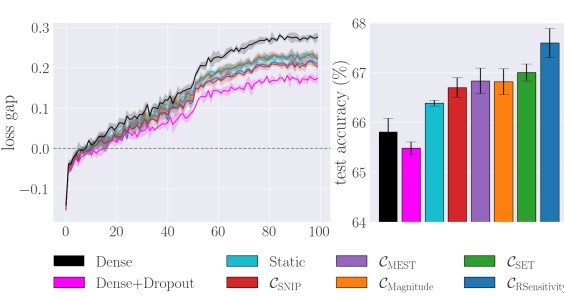

Figure 4: **Left:** The loss gap (validation loss $-$ training loss) over time for different pruning criteria, the static sparse model, the dense model, and the dense model with dropout $p = 0.05$ on the large-MLP. **Right:** The test accuracy obtained for each criterion.

Within each studied density, growing criterion, and model, we rank the pruning criteria and then calculate their average ranks. To rigorously establish the statistical significance of our findings, we compute the Critical Distance Diagram for those ranks, using the Nemenyi post-hoc test [10] with p-value $0.05$ and present it in Figure 3. We observe that in the low-density regime (Figure 3b), the $\mathcal{C}_{\text{Magnitude}}$ criterion achieves the best performance, as given by the lowest average rank. Furthermore, in such a case, we also note that other predominately magnitude-based criteria, such as $\mathcal{C}_{\text{SET}}$ and $\mathcal{C}_{\text{MEST}}$ are not significantly different than $\mathcal{C}_{\text{Magnitude}}$, while the remaining gradient-based approaches are clearly worse. This confirms the overall effectiveness of the

---

[6]We study the impact of the $\lambda$ hyperparameter of the $\mathcal{C}_{\text{MEST}}$ criterion in Appendix A.3. In general, we see that $\lambda \to 0$ leads to results equal to $\mathcal{C}_{\text{Magnitude}}$, while $\lambda \to \infty$ prioritizes only the gradient magnitude and degrades the performance. We do not see any clear improvement for values lying between those two extremes.

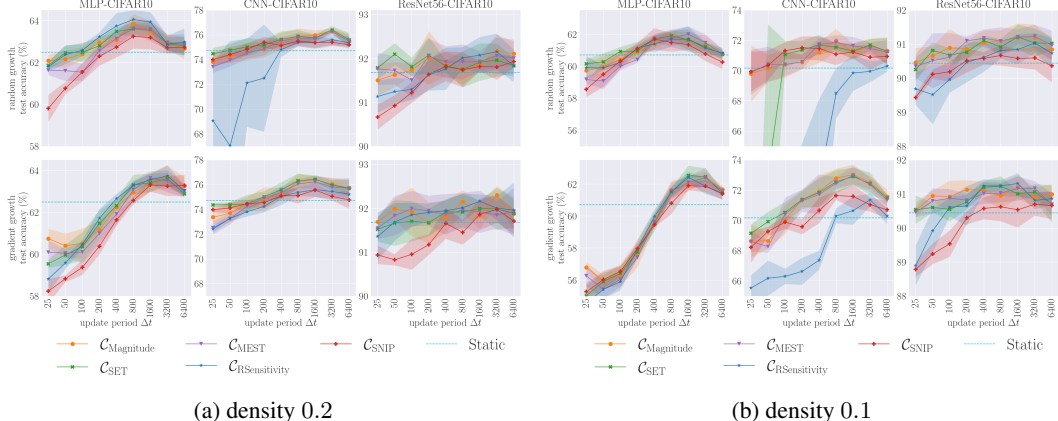

(a) density 0.2                                         (b) density 0.1

Figure 5: The update period $\Delta t$ versus validation accuracy on the CIFAR10 dataset on the MLP, ConvNet, and ResNet-56 models for different pruning criteria with density 0.2 and 0.1. The top row corresponds to random growth, while the bottom row corresponds to gradient growth. Note the logarithmic scale on the x-axis. We see that the methods are most sensitive to the update frequency in the MLP setting. For all setups, performing the update later ($\Delta t > 400$) seems to be beneficial.

weight's magnitude as an importance score in sparse regimes. See also Appendix G for the discussion of the effect of batch size on the obtained results. Interestingly, when larger densities are involved, the best choice is given by $\mathcal{C}_{\text{RSensitivity}}$. Additionally, the average rank of $\mathcal{C}_{\text{magnitude}}$ deteriorates. This may suggest that the information from the gradient is more reliable in such a case. The gradient-based methods seem to perform better in denser setups.

Finally, we notice that in certain situations, the DST framework can lead to better results than the original dense model (see small-MLP, or ResNet56 on CIFAR100 in Figure 1). We argue that this is due to a mix of a regularization effect and increased expressiveness introduced by removing and adding new connections in the network – see Figure 4. Moreover, we also observe that within the setup studied in Figure 1, there is almost no difference between choosing the random or gradient *growth criterion*. The second one achieves slightly better accuracy for the convolutional models (see, e.g., density 0.2 on ResNet-56, as well as improved stability for EfficientNet). The similarity between those two growth criteria and the surprising effectiveness of DST in comparison to dense models have also been previously observed in the context of Reinforcement Learning [18].

## 5.3 Sensitivity to Update Period

In this section, we analyze the sensitivity of each pruning criterion to the update period $\Delta t$. This value describes how often the adjustments are made in training and hence can affect the weights' importance scores. We fix the density to 0.2 and 0.1 and the initial pruning fraction $\rho$ to 0.5. We choose the densities 0.2 and 0.1 as they correspond to rather high sparsity levels and generally provide reasonable performance. Next, we investigate different update periods on the models using CIFAR10 dataset.[7] We start from $\Delta t = 25$ and increase it by a factor of 2 up to $\Delta t = 6400$.

The results are presented in Figure 5. For the MLP model, we clearly see the behavior described in Section 4: if the update period $\Delta t$ is too small, the performance will suffer. The best update period setting seems to be $\Delta t = 800$ for the random growth, regardless of the pruning method. For gradient growth, this value also gives a good performance. However, larger pruning periods (e.g. $\Delta t = 1600$) are even better.[8] Note that we use a batch size of 128, hence for $\Delta t = 800$ the connectivity is changed approximately once per 2 epochs.[9] Similarly, less frequent updates are also beneficial in the ResNet56 model, although there is not as much consensus between the different pruning criteria. Even performing just one topology update every $\Delta t = 6400$ iterations (approximately once per 16 epochs) still gives excellent performance. At the same time, not performing any topology updates at

---

[7]The update period is only meaningful when considered together with the size of the training set. Therefore it is important to evaluate models of different sizes on the same dataset.

[8]See also Appendix J for an analysis of the effect of batch size and pruning schedule on these results.

[9]Since $800 \cdot 128 = 102,400$ samples, which is just over twice the training set size of CIFAR10 (50,000).

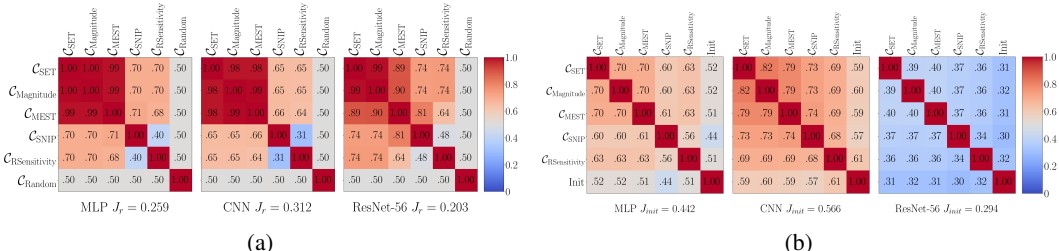

Figure 6: **(a)** The mean normalized Jaccard index between the sets of weights chosen for removal during the first update of the sparse connectivity, computed for the MLP, CNN, and ResNet-56 models on CIFAR10. The $J_r$ indicates the expected overlap of random subsets and serves as a reference point. We estimate this value by computing the mean pair-wise Jaccard index of the random pruning criterion with different sparse initialization. **(b)** The same index computed between the masks obtained by different pruning criteria at the end of training. The rows and columns represent the pruning criteria between which the index is computed. The $J_{init}$ denotes the expected overlap of random masks and serves as a reference point. We estimate this value by computing the mean pair-wise Jaccard index of the different sparse initializations.

all (i.e., static sparse training) deteriorates performance, as shown by the cyan dashed line in Figure 5. Furthermore, we observe that the gradient-based pruning criteria seem to suffer much more than the magnitude criteria when paired with a small update period (consider the plots for $\Delta t = 25$), especially when combined with the gradient growth.[10] This suggests that it is safer not to use those techniques together or be more rigorous about finding the right update period in such a case.

Most interestingly, the results from Figure 5 strongly suggest that frequent connectivity adjustment is not needed. Even few pruning and re-growing iterations are enough to improve the static initialization.

### 5.4 Structural Similarity

Finally, we investigate whether the studied criteria are diverse in terms of the selected weights for removal. We fix the dataset to CIFAR10 and analyze the moment in which the sparse connectivity is updated for the first time. At such a point, given the same sparse initialization and random seed, all the pruning criteria share the same network state. Consequently, we can analyze the difference between the sets selected for removal by each criterion. We do this by computing their Jaccard index (Equation 1) and present the averaged results of 5 different runs in Figure 6a (See also Appendix K standard deviations and more plots).

We observe that the $\mathcal{C}_{\text{SET}}$ and $\mathcal{C}_{\text{Magnitude}}$ mainly select the same weights for pruning. This indicates that the absolute values of the positive and negative weights are similar. In addition, for the simpler models with smaller depths (MLP and CNN) the $\mathcal{C}_{\text{MEST}}$ criterion also leads to almost the same masks as $\mathcal{C}_{\text{Magnitude}}$. On the other hand, the $\mathcal{C}_{\text{SNIP}}$ and $\mathcal{C}_{\text{RSensitivity}}$ criteria produce sets that are distinct from each other. This is natural, as $\mathcal{C}_{\text{SNIP}}$ multiplies the score by the magnitude of the gradient, while $\mathcal{C}_{\text{RSensitivity}}$ uses its inverse. The experiment suggests that early in training, pruning criteria such as $\mathcal{C}_{\text{Magnitude}}$, $\mathcal{C}_{\text{SET}}$, and $\mathcal{C}_{\text{MEST}}$ lead to similar sparse connectivity. Together with the similarity in performance observed in Section 5.2, the results indicate that these three methods are almost identical, despite often being presented as diverse in DST literature. At the same time, the updates made by $\mathcal{C}_{\text{SNIP}}$ and $\mathcal{C}_{\text{RSensitivity}}$ produce different pruning sets, and result in lower performance (recall Figure 3b). However, note that for each entry (except $\mathcal{C}_{\text{SNIP}}$ with $\mathcal{C}_{\text{RSensitivity}}$), the computed overlap is larger than random, suggesting that there is a subset of weights considered important by all the criteria.

In addition, we also compare how similar are the final sparse solutions found by the methods. To this end, we fix the sparse initialization to be the same for each pair of criteria and compute the Jaccard Index of the masks at the end of the DST training (performed with gradient growth and density 0.2 on the CIFAR10 experiments from Section 5.2). For each pair, we average the score over 5 initializations. The resulting similarity matrix is presented in Figure 6b. We observe that pruning criteria that selected similar sets of weights for removal from the previous experiment still hold some resemblance to each other in terms of the end masks. Let us also note that in Section 5.3, we observed

---

[10]In Appendix E we also study the exploration imposed by the pruning criteria using the ITOP ratio [38].

that applying just a few connectivity updates is sufficient for good results. This raises the natural question of whether DST masks are genuinely diverse from their sparse initializations. By analyzing the bottom row in Figure 6b, we see that the masks obtained at the end of the DST training are distinct from their corresponding initializations, having the Jaccard Index close to that of a random overlap.

In consequence, we conclude that the best-performing methods from Section 5.2 indeed make similar decision choices while having a smaller overlap with the less efficient criteria. This renders the magnitude-based criteria almost equivalent, despite their separate introduction in the literature. At the same time, the DST mask updates made during the training are necessary to adapt and outperform the static initialization. We would like to highlight that such insights could not have been derived solely from performance and hyperparameter results. We hope that our insights will raise awareness of the importance of performing structural similarity experiments in the DST community.

## 6 Conclusion

We design and perform a large study of the different pruning criteria in dynamic sparse training. We unveil and discuss the complex relations between these criteria and the typical DST hyperparameters settings for various models. Our results suggest that, overall, the differences between the multiple criteria are minor. For very low densities, which are the core of DST interest, criteria based on magnitude perform the best. This questions the effectiveness of gradient-based scores for pruning during the training. We propose to incorporate the selected models and datasets as a baseline in future works investigating the pruning criteria for DST to avoid the common case where methods are overfitted to given DST hyperparameters or tasks. We hope that our research will contribute to the understanding of the sparse training methods.

**Limitations & Future Work.** Our research was conducted mainly on datasets from computer vision and one tabular dataset. It would be interesting to verify how our findings translate to large language models. The computed structural insights are based on set similarity and disregard the information about the value of the parameters. Additional topographic insights could be provided to incorporate this information and analyze the graph structure of the found connectivity. Finally, much work still needs to be done on the hardware side to truly speed up training times and decrease memory requirements (see Appendix F). We do not see any direct social or ethical broader impact of our work.

## Acknowledgements

The research of Aleksandra Nowak and Jacek Tabor has been supported by the flagship project entitled *Artificial Intelligence Computing Center Core Facility* from the DigiWorld Priority Research Area under the Strategic Programme Excellence Initiative at Jagiellonian University. The work of J. Tabor was supported by the National Centre of Science (Poland) Grant No. 2021/41/B/ST6/01370. The research of Bram Grooten was funded by the project AMADeuS (with project number 18489) of the Open Technology Programme, which is partly financed by the Dutch Research Council (NWO). We gratefully acknowledge Polish high-performance computing infrastructure PLGrid (HPC Centers: ACK Cyfronet AGH) for providing computer facilities and support within computational grant no. PLG/2022/015887.

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

# A  Experiments Setup

## A.1  Model Architectures

For the small-MLP setup, we use an MLP with 2 hidden layers with dimension sizes of 256. For the large MLP, we increase the number of neurons in the first hidden layer to 1024, and to 512 in the second. The small-CNN is a standard ConvNet architecture consisting of three convolutional layers with kernel size 3, stride 1, and padding 1, inspired by the standard ConvNet presented in section 8.2.3 of [7]. The spatial dimension reduction is implemented by MaxPooling with size 2. In all cases, we use ReLU activations. The architectures are summed up in Table 2. The ResNet-56 and ResNet-50 models follow their standard specifications [22]. The EfficientNet is the EfficientNet-B0 model [57] (we use the implementation from `pytorch` models), and follows the standard architecture. The VGG-16 model is modified to be used with CIFAR-10 and uses BatchNorm. The implementation of VGG-16, as well as the LeNet-5-Caffe model, is adapted from [34].

Table 2: The small-MLP, large-MLP, and small-CNN architectures used in this study. The first two pairs next to a Linear layer indicate the input and output dimensions. The tuple $(c_{in}, c_{out}, k, s)$ after the Conv2d described the input channels, output channels, kernel size, and stride, respectively. The ResNet-50, and ResNet-56 architectures follow the standard specifications [22] (25,502,912 and 855,770 parameters, respectively). All networks use a Softmax nonlinearity after the last mentioned layer.

| Layer | small-MLP | large-MLP | small-CNN |
|---|---|---|---|
| 1 | $\text{Linear}(24, 256)$ | $\text{Linear}(3*32*32, 1024)$ | $\text{Conv2d}(3, 32, 3, 1)$ |
| 2 | ReLU | ReLU | ReLU |
| 3 | $\text{Linear}(256, 256)$ | $\text{Linear}(1024, 512)$ | MaxPool, size 2 |
| 4 | ReLU | ReLU | $\text{Conv2d}(32, 64, 3, 1)$ |
| 5 | $\text{Linear}(256, 1)$ | $\text{Linear}(512, 10)$ | ReLU |
| 6 | | | MaxPool, size 2 |
| 7 | | | $\text{Conv2d}(64, 128, 3, 1)$ |
| 8 | | | ReLU |
| 9 | | | Global Average Pool |
| 10 | | | $\text{Linear}(128, 10)$ |
| Num. Parameters (Dense) | 72,449 | 3,676,682 | 94,538 |
| Num. Params. (95% Sparse) | 3,622 | 183,834 | 4,727 |

## A.2  Training Regime

We provide an overview of the datasets we used in our experiments is shown in Table 4. Below, we report the training regime.

**Small- and large- MLPs** We use a batch size of 128, the SGD optimizer with momentum 0.9, weight decay 0.0005, and Nesterov= `True`. The learning rate starts at 0.01 and is decayed twice during training: after 50% and 75% of the epochs. We use the decay of 0.1. We train for 100 epochs.

**Small-CNN** We use a batch size of 128, the SGD optimizer with momentum 0.9, weight decay 0.0005, and Nesterov= `True`. The learning rate starts at 0.01 and is decayed twice during training: after 50% and 75% of the epochs. We use the decay of 0.1. We train for 100 epochs.

**LeNet-5-Caffe** We use a batch size of 128, the SGD optimizer with momentum 0.9, weight decay 0.0005, and Nesterov= `True`. The learning rate starts at 0.01 and is decayed twice during training: after 50% and 75% of the epochs. We use the decay of 0.1. We train for 100 epochs.

**ResNet-56** We use a batch size of 128, the SGD optimizer with momentum 0.9, weight decay 0.0001, and Nesterov= `True`. The learning rate starts at 0.1 and is decayed twice during training: after 50% and 75% of the epochs. We use the decay of 0.1. We train for 200 epochs.

**ResNet-50** We use a batch size of 256, the SGD optimizer with momentum 0.9, weight decay 0.0001, and Nesterov= `True`. The learning rate starts at 0.1 and is decayed twice during training: after 50% and 75% of the epochs. We use the decay of 0.1. We train for 90 epochs.

**VGG-16** We use a batch size of 128, the SGD optimizer with momentum 0.9, weight decay 0.0001, and Nesterov= `True`. The learning rate starts at 0.01 and is decayed twice during training: after 50% and 75% of the epochs. We use the decay of 0.1. We train for 200 epochs.

**EfficientNet** We use a batch size of 128, the SGD optimizer with momentum 0.9, weight decay 0.0001, and Nesterov= `True`. The learning rate starts at 0.01 and is decayed twice during training: after 50% and 75% of the epochs. We use the decay of 0.001. We train for 100 epochs. We train on the 64x64 Tiny-ImageNet images (we do not re-scale them to 224x224 and we do not use pre-training in the main text experiments).

**DST** Settings specific for dynamic sparse training: We use the ER initialization for the distribution of sparsity levels in MLP, and the ERK initialization for the convolutional models. We start with an initial *pruning fraction* $\rho$ of 0.5, which is decayed during training using a cosine decay schedule (see the main text). The default topology update period $\Delta t$ is set to 800, but in the experiments for **Q2** we vary this hyperparameter. Note that we also keep the update period $\Delta t = 800$ for the ImageNet, since together with batch size 256 it matches the number of training examples seen by the model before the next topology update reported in [12]. For completeness, we also provide the DST hyperparameter setups used to obtain the Figures in the main text in Table 3.

Table 3: The DST hyperparameters used to obtain the results presented in the main-text experiments.

| Experiment | Prune Fraction | Prune Fraction Decay Scheduler | Density | Update Period |
|---|---|---|---|---|
| Fig. 1 | 0.5 | cosine | 0.05, 0.06, 0.07, 0.08, 0.09, 0.1, 0.15, 0.2, 0.3, 0.4, 0.5 | 800 |
| Fig. 2 | 0.5 | cosine | 0.2 | 800 |
| Fig. 4 | 0.5 | cosine | 0.2, 0.1 | 25, 50, 100, 200, 400, 800, 1600,3200,6400 |
| Fig. 5 | 0.5 | cosine | 0.5 | 800 |
| Fig. 6a | 0.5 | cosine | 1.0 | 800 (measured only once) |
| Fig. 6b | 0.5 | cosine | 0.2 | 800 |

**Other Setup and Infrastructure Details** We perform 5 runs for each setup, except for the ImageNet dataset, in which we use 3 runs. We do not fine-tune anything specific for any model. In all the experiments performed, at the beginning of the code, we fixed all the random seeds (`random.seed`, `numpy.random`, `torch.manual_seed`, `torch.cuda.manual_seed`), and set the `cudnn` backend to `deterministic`. The DST pruning criteria (if not used with random growth) are deterministic and for the same seed start with the same sparse initialization. While computing the Jaccard Index between the end masks we use only one worker for data-loading. In terms of reproducibility, we provide the environment setup but also perform our experiments in a Singularity container, for which we will provide the definition file in the repository. We provide the hyperparameter setups for all the main-text experiments in the code.[11]

We used GPUs either of the NVIDIA Tesla V100 or NVIDIA A100 type. We used external servers with job scheduling. All experiments (except for ImageNet) were performed on a single GPU. No other computations could access that GPU at such time. The ImageNet experiments were run using distributed computing on eight NVIDIA A100. Each task was using a batch size of 32 (so that the total batch size is 256) and the hyperparameter configuration described for the ResNet50 above.

Our code is accessible at `https://github.com/alooow/fantastic_weights_paper`. It is based on the publicly available repositories for [34] and [11] available at `https://github.com/VITA-Group/Random_Pruning` and `https://github.com/TimDettmers/sparse_learning`, respectively. The used sparse-learning library (of [11]) is published under the MIT license.[12]

---

[11]Under the `specs` directory in the code.

[12]We include the copy of that licence in the `sparse-learning` directory in our code.

Table 4: Datasets overview. The validation and training set on ImageNet follow the standard division. For CIFAR-10, we use 5000 of the training examples to form the validation test and fit the models on the remaining 45000. The test set for CIFAR-10 follows the standard division.

| dataset | type | samples | features | classes | train-valid-test | source | URL |
|---|---|---|---|---|---|---|---|
| Higgs | tabular | 940,160 | 24 | 2 | 70%-15%-15% | [2] | openml.org/d/44129 |
| CIFAR10 | image | 60,000 | 3x32x32 | 10 | 75%-8%-17%[13] | [27] | cs.toronto.edu/~kriz/cifar.html |
| CIFAR100 | image | 60,000 | 3x32x32 | 100 | 75%-8%-17%[14] | [27] | cs.toronto.edu/~kriz/cifar.html |
| ImageNet | image | 1,431,167 | 3x469x387[15] | 1000 | 90%-3%-7%[16] | [49] | image-net.org/challenges/LSVRC/2012/ |
| FashionMNIST | image | 70,000 | 1x28x28 | 10 | 77%-9%-14%[17] | [63] | github.com/zalandoresearch/fashion-mnist |
| Tiny-ImageNet | image | 100000 | 3x64x64 | 200 | 80%-10%-10%[18] | [29] | cs231n.stanford.edu/tiny-imagenet-200.zip |

### A.3 Tuning MEST

Recall from Section 3.2 that the MEST criterion is computed as $s(\theta) = |\theta| + \lambda|\nabla_\theta \mathcal{L}(\mathcal{D})|$. Since $\mathcal{C}_{\text{MEST}}$ depends on the $\lambda$ hyperparameter, we verify how the change of that value impacts the overall results for various densities. We present the results on the large-MLP model in Figure 7. We observe that for $\lambda$ close to $0$, the MEST criterion is, as expected, close to the magnitude criterion. Increasing $\lambda$ does not seem to benefit the performance. In particular, depending highly on the gradient is disadvantageous.

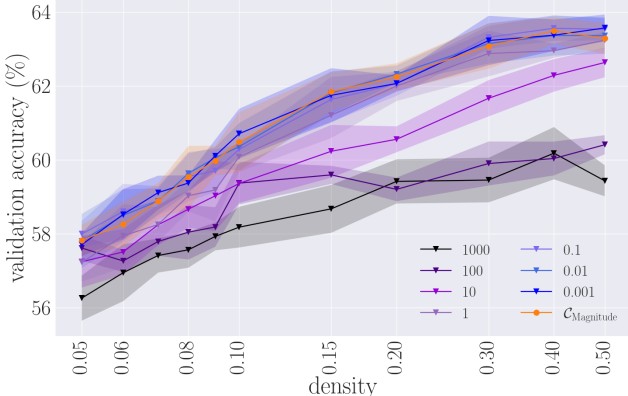

Figure 7: The performance of the MEST criterion with random growth for different parameters of the hyperparameters $\lambda$. We observe that for low values of $\lambda$, the criterion behaves similarly to magnitude, while for very large values, the performance drops. There does not seem to be a good value between those two extremes.

## B  Sensitivity and Reciprocal Sensitivity

We investigate the relation between the Sensitivity and Reciprocal Sensitivity on the large-MLP model on CIFAR10. The results are presented in Figure 8. We observe that not only is the Reciprocal criterion better in terms of the attained test accuracy, but it also seems to be more stable throughout the training. We argue that this is due to how those two criteria deal with weights that are large in magnitude but have small gradients. Note that such weights can be considered stable and hence

---

[13]These are rounded percentages. We use the standard split of 45,000 - 5,000 - 10,000 images.

[14]These are rounded percentages. We use the standard split of 45,000 - 5,000 - 10,000 images.

[15]This is the average resolution. Images are randomly cropped to size 3x224x224, as is standard practice.

[16]Percentages are rounded. We use the standard train/validation/test split. We do not use the test set and evaluate on the validation set.

[17]Percentages are rounded. We apply the default train/test split (60000/70000) and move a random 10% of the train set as a validation set.

[18]We use the standard train/validation/test split. We use the validation set as the test set (we *do not* optimize any hyperparameters using the validation set).

be desired to keep active. While the Reciprocal criterion would assign them a very high score, the Sensitivity, having an inverse relation to the gradient, would most likely mark them for removal.

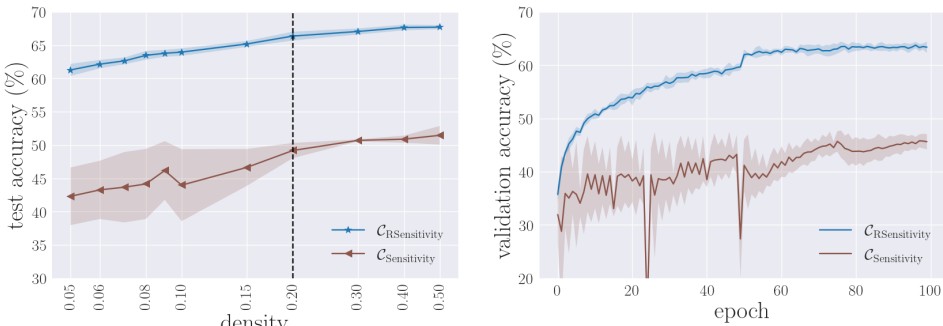

Figure 8: **Left**: The density versus test accuracy for the MLP dataset on CIFAR10 obtained by the Reciprocal Sensitivity and Sensitivity pruning criteria. **Right**: The validation accuracy throughout the training for density 0.2 (denoted by a vertical dashed line in the first plot). We may observe that the Reciprocal Sensitivity is more stable and gives much better results.

## C    Results On Text Data

In addition to the tasks studied in the main text, we conduct an evaluation of the pruning criteria on the fine-tuning task of ROBERTa Large [39] with approximately 354 million parameters, utilizing the CommonsenseQA dataset [56] adapted from the Sparsity May Cry (SMC) Benchmark [35].

We use the exact same setup as in the SMC-Benchmark by first performing magnitude pruning for the sparse initialization and then using DST during the fine-tuning phase. We also use the same hyperparameters. Please note that this is a different configuration from the one used in the main text, in which we trained all the models from scratch (in spirit, the CommonsenseQA study is more similar to the EfficientNet fine-tuning experiment from Appendix H). We report the results in Table 5 and Table 6 for the random and gradient growth, respectively.

Similar to the findings of [35], we observe that in this scenario, the achievable sparsity without a significant performance decline is notably lower than in vision or tabular data scenarios. Concerning random growth, the $\mathcal{C}_{MEST}$ criterion appears to be the most suitable choice. For gradient growth with a density of 0.8, the $\mathcal{C}_{RSensitivity}$ and $\mathcal{C}_{SET}$ criteria initially demonstrate strong performance but are eventually surpassed by the $\mathcal{C}_{magnitude}$ and $\mathcal{C}_{MEST}$ criteria at a density of 0.7. Additionally, we note a significantly higher variance in the outcomes across all criteria compared to vision tasks.

Furthermore, we conducted a brief exploration of update period values for this problem (refer to Figure 9). Notably, we observe that overly frequent updates do not yield beneficial results. Concerning gradient growth, the most effective update period value is consistently around $\Delta t = 500$ for nearly all criteria. For random growth higher update period values generally lead to improved outcomes.

At the same time, please note that the studied pruning criteria had been evaluated either on tabular or vision datasets in most of the works that have introduced them or used them (e.g., [42, 64, 45, 12, 11]). Large language models typically have a different structure and are based on the attention mechanism. In consequence, the applicability of sparse training in such a setup is still an open area of research [35, 55]. This motivated our choice of fixing our focus on tabular and vision models in the main text.

## D    Additional Performance Plots

In this section we present the performance of the different pruning criteria on the two remaining architectures from the main text - the small-CNN trained on CIFAR10, and ResNet56 trained on CIFAR100. The results are in Figure 10 (see also Figure 22 for all the models in one plot).

Table 5: The mean accuracy results for the CommonsenseQA task on ROBERTa Large Model from the SMC-Benchmark (Liu et al. 2023). We use the same hyperparameter setup as in (Liu et al. 2023) and vary only the density and pruning criterion. The results below were obtained for random growth. In brackets, we report the standard deviation from 3 runs. We bold out the best result for each density and underline the second-best one.

| | random growth | | | | |
| | 0.9 | 0.8 | 0.7 | 0.6 | 0.5 |
| --- | --- | --- | --- | --- | --- |
| $\mathcal{C}_{magnitude}$ | 73.80 (0.52) | 37.57 (25.25) | 26.53 (6.71) | 19.80 (3.13) | 20.60 (0.69) |
| $\mathcal{C}_{SET}$ | 74.40 (0.60) | 49.50 (26.98) | 21.00 (3.99) | 20.13 (1.95) | 18.63 (1.44) |
| $\mathcal{C}_{MEST}$ | **74.77 (0.67)** | **52.30 (29.79)** | **26.90 (7.20)** | 19.77 (0.49) | 20.73 (0.96) |
| $\mathcal{C}_{RSensitivity}$ | 25.87 (8.36) | 22.13 (1.69) | 23.27 (0.85) | 21.37 (1.66) | **21.20 (0.53)** |
| $\mathcal{C}_{SNIP}$ | 56.57 (1.82) | 34.20 (9.11) | 19.57 (1.31) | **21.60 (0.95)** | 19.17 (1.47) |

Table 6: The mean accuracy results for the CommonsenseQA task on ROBERTa Large Model from the SMC-Benchmark (Liu et al. 2023). We use the same hyperparameter setup as in (Liu et al. 2023) and vary only the density and pruning criterion. The results below were obtained for gradient growth. In brackets, we report the standard deviation from 3 runs. We bold out the best result for each density and underline the second-best one.

| | gradient growth | | | | |
| | 0.9 | 0.8 | 0.7 | 0.6 | 0.5 |
| --- | --- | --- | --- | --- | --- |
| $\mathcal{C}_{magnitude}$ | 76.00 (0.46) | 55.80 (29.51) | **37.37 (18.23)** | **24.27 (2.94)** | 19.33 (0.35) |
| $\mathcal{C}_{SET}$ | **76.30 (0.75)** | 68.43 (6.15) | 28.57 (1.85) | 22.60 (1.47) | **22.17 (0.93)** |
| $\mathcal{C}_{MEST}$ | 74.73 (0.15) | 53.90 (29.27) | 37.53 (8.45) | **24.27 (2.21)** | 19.63 (1.12) |
| $\mathcal{C}_{RSensitivity}$ | 75.83 (0.55) | **69.60 (6.70)** | 25.43 (5.65) | 20.67 (2.81) | 21.57 (1.82) |
| $\mathcal{C}_{SNIP}$ | 65.50 (2.27) | 48.20 (7.79) | 32.70 (5.98) | 23.17 (4.82) | 20.73 (0.61) |

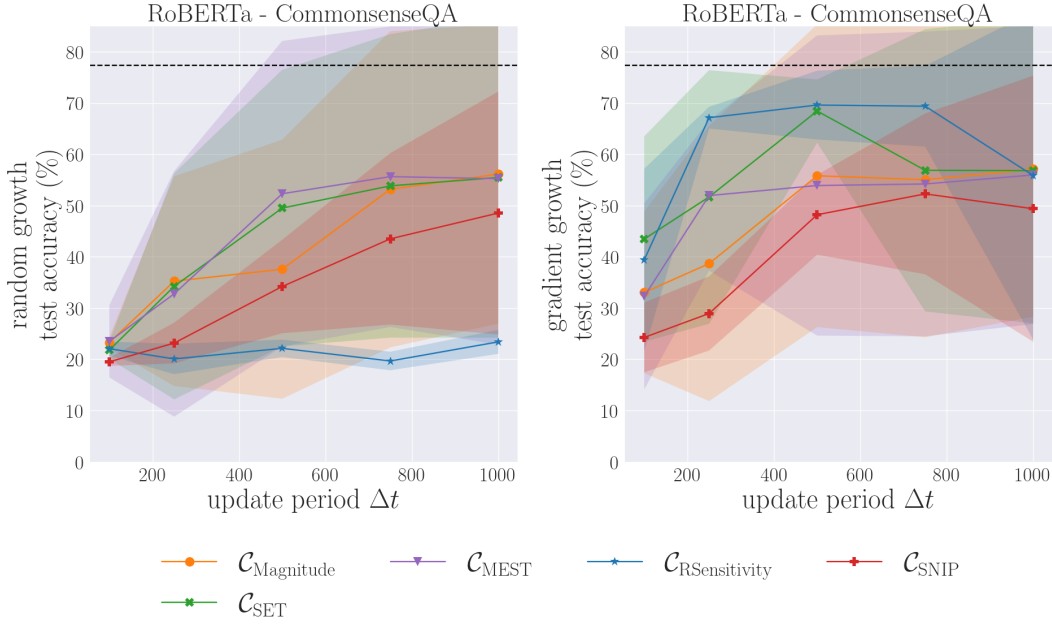

Figure 9: The study of the effect of the change of update period $\Delta t$ on the performance in the CommonsenseQA task on ROBERTa Large. Left: random growth, right: gradient growth.

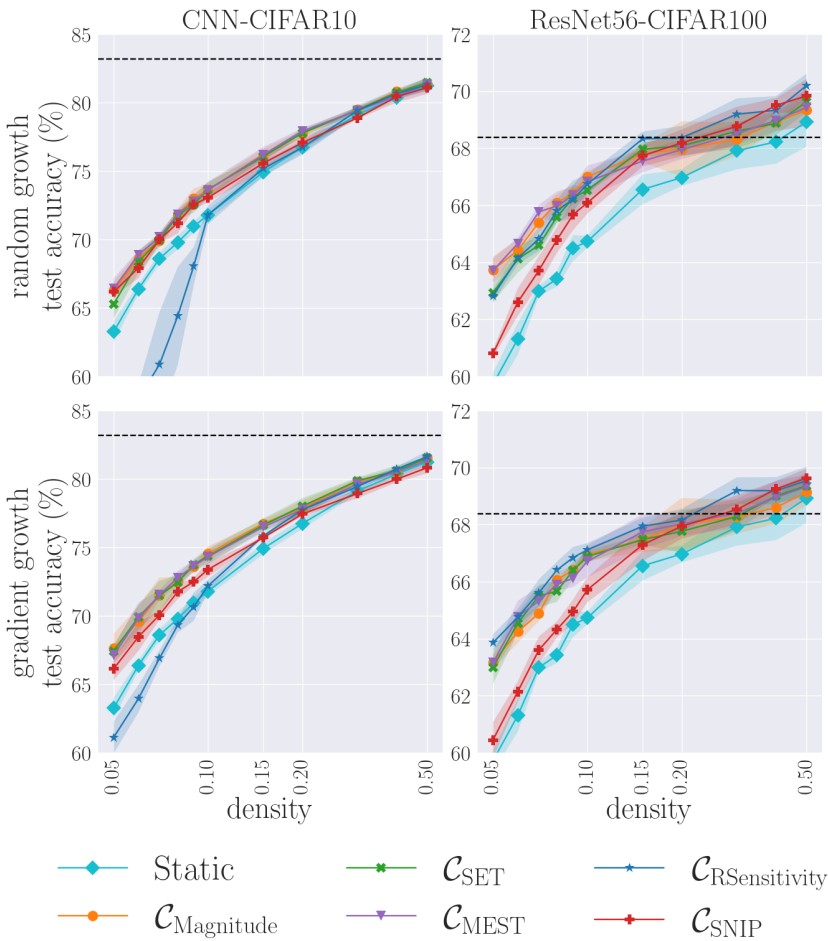

Figure 10: The performance of the pruning criteria on the CNN-CIFAR10 and ResNet56-CIFAR100 datasets for different densities. Please note the logarithmic scale on the x-axis. We observe that the differences between the methods are larger for the low densities.

We observe that again the differences between the studied methods seem to be more visible in the low-density regime. In that setup, the $\mathcal{C}_{SNIP}$ criterion is achieving worse test accuracy than the $\mathcal{C}_{MEST}$ and the magnitude-based methods. The $\mathcal{C}_{RSensitivity}$ also performs poorly on the CNN-CIFAR10 model. We can see that the behavior for the VGG-16 network observed in the plots in the main text is not due to the change of dataset, but rather the change of the model (since ResNet56 on CIFAR100 has similar results to ResNet56 on CIFAR10 from the main text).

## E In-Time-Over-Parametrization (ITOP) Study

A known component that has been used to explain sparse training performance together with update-period sensitivity is the ITOP (In Time Over-Parametrization) ratio [38]. The ITOP measures the number of 'explored' parameters divided by the total number of parameters. The parameter is 'explored' if it has been included in the mask at least once during the whole training.

We study the relationship between ITOP, update period, and the pruning criteria on the example of the large-MLP model on CIFAR10 with density 0.1. The results are shown in Figure 11. There are some interesting insights gained by this analysis.

First, we observe that when paired with random growth, the choice of pruning criteria leads to almost identical final ITOP ratios. (Note that the points for different pruning criteria overlap in the left plot, yet we verify that they are not exactly equal). This is expected because, in such a case, the growth

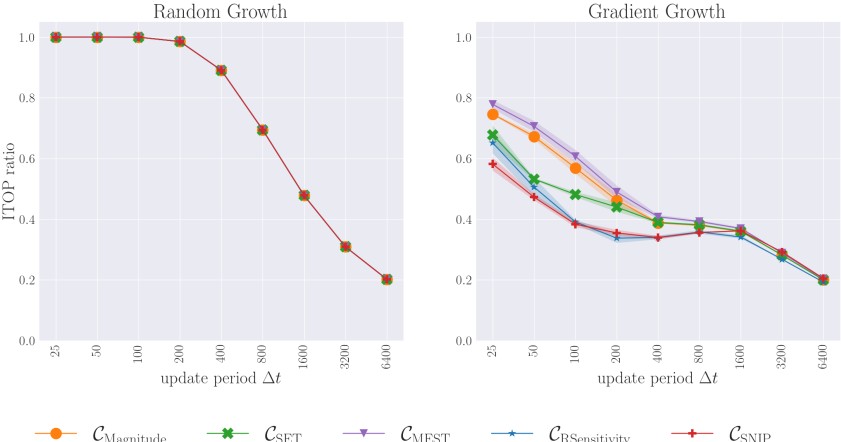

Figure 11: The update period $\Delta t$ versus the ITOP ratio obtained at the end of the training for different pruning criteria on the large MLP. The left plot corresponds to random growth, the right plot corresponds to gradient growth.

criterion is independent of the chosen set of weights to prune. Hence, the increase in ITOP is only due to the growth criterion, which (being random selection) is not affected by the pruning criterion. This is in contrast to gradient growth, in which the change in the pruned weights can impose a change to the gradient flow in the model and hence impact the rate of the exploration. Aside from this observation, we confirm that gradient growth has, in general, a lower ITOP ratio than random (which was shown in [38]).

Secondly, in the gradient growth setup, the magnitude criteria and $\mathcal{C}_{MEST}$ have a much higher ITOP ratio for small update periods than the gradient-based pruning criteria. This indicates that they perform more exploration, which perhaps may be the reason behind their better performance observed in Figure 4 in the main text.

Finally, we notice that final ITOP ratio close to $1$ is not necessary for good generalization. Observe that for $\Delta t = 800$, we get the best validation accuracy for random growth, MLP, and CIFAR10 (Figure 5 in our paper), but it only reaches an ITOP ratio of $0.7$. Even with a final ITOP ratio of just $0.2$ (for $\Delta t = 6400$), sparse models can beat the performance of dense models.

## F  Sparse Hardware and Software Support

As discussed in [53], research on sparsity is being pursued in three main areas to create faster, more efficient neural networks. The first area is the development of hardware that can take advantage of sparse networks, such as NVIDIA's A100, which supports 2:4 sparsity (a restricted version of 50% layer sparsity) [65]. The second area is the creation of software libraries that can implement truly sparse networks [37, 8]. The third area is the development of algorithms that can produce sparse networks that perform just as well as dense networks [23]. All these steps are being undertaken as a collaborative community effort, aiming to create faster, memory- and energy-efficient deep neural networks. Additional information on this topic can be found in [24, 41].

## G  Gradient Based Methods and Batch Size

In Section 5.2 in the main text, we observe that, in general, the magnitude-based pruning criteria perform better in sparse regimes than the gradient-based ones. Note that the gradient is computed for a batch (using SGD), and hence, small batch sizes might introduce more significant errors in Taylor-approximation based pruning (e.g., SNIP). At the same time, computing the gradient over the entire training set is not feasible in DST since it would significantly increase the time complexity of each mask update.

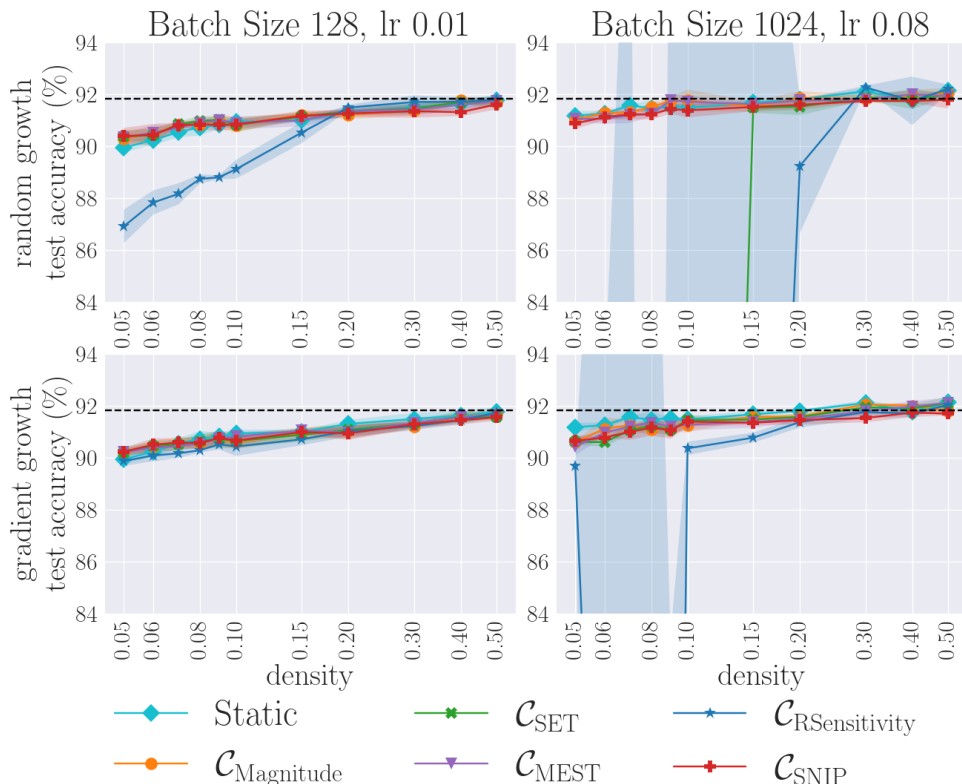

Figure 12: The performance of different pruning criteria on LeNet-5-Caffe on FashionMNIST. The first row corresponds to random growth, the second row corresponds to gradient growth. The left column indicates a batch of size 128, and the right column indicates a batch of size 1024. The black dashed line in both plots is the performance of the dense model from the main text.

To investigate the impact of the batch size on our results, we perform a small study on the LeNet-5-Caffe model (trained on FashionMNIST), in which we increase the batch size almost 10 times up to 1024. At the same time, we increase the learning rate up to 0.08 from 0.01 so that the learning-rate-to-batch-size ratio stays approximately the same (as suggested by the scaling laws, for instance, in [51]). We compare the obtained results with the standard setup with a batch size of 128. We also decrease the update period from 800 to 100 so that in both setups, the DST updates are made after seeing the same number of training examples. We report the results in Figure 12.

Interestingly, we observe that for batch size 1024 $\mathcal{C}_{SNIP}$ indeed seems to perform slightly better. $\mathcal{C}_{RSensitivity}$ and $\mathcal{C}_{SET}$, on the other hand, become significantly unstable, and $\mathcal{C}_{MEST}$ is still indistinguishable from magnitude pruning. Those results may indicate that the impact of the batch size, although existent, is not significant. Furthermore, as we will see in the next paragraph, increasing only the batch size of the DST update is also not beneficial.

**Increasing Only the Batch Size of the DST update.** In addition to the above-conducted experiments, we examine an ablation in which we increase the batch size to 1024 only when computing the gradient used for the DST update. During normal steps of optimization, we use the standard choice from previous experiments (128). The results are in Figure 13. We observe that it does not enhance performance. Let us note, however, that, in general, the impact of the batch size still needs additional investigation in larger models and datasets in order to be evaluated with certainty.

## H  Fine-Tuning EfficientNet and the Choice of Pruning Criterion

In all experiments in the main text, we perform the training from scratch (no pretraining is applied). This includes the EfficientNet on the Tiny-ImageNet from the main text. In this section, we also analyze the difference in the performance of the pruning criteria when evaluated in a fine-tuning

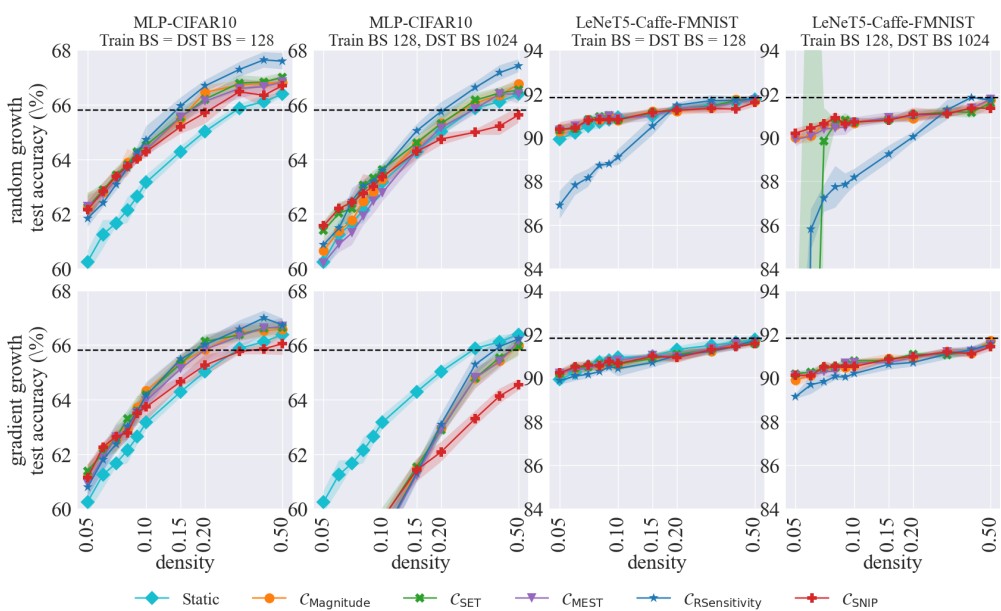

Figure 13: The performance of MLP-CIFAR10 and LeNet5-Caffe-FashionMNIST from the main text experiments (first and third column) and the performance when using an increased batch *only* for the DST update (second and last column).

scenario. In such a setup, we initialize the weights of the EfficientNet to use the pretrained parameters from ImageNet. We only re-initialize the linear projection in the head of the model (which needs to have the output dimension changed from 1000 to 200 since there are 200 classes in Tiny-ImageNet). We keep all hyperparameters the same as in the training-from-scratch setup (see Appendix A). We compare the results in Figure 14

We observe that in the setup that uses the pretrained weights, the $\mathcal{C}_{\mathcal{SNIP}}$ criterion performs much better than when training from scratch. We hypothesize that perhaps the use of the Taylor approximation of the change in the Loss function (used in the SNIP criterion) allows not to diverge far from the already quite well-established solution from the pretraining. The purely magnitude-based methods, on the other hand, disregard the potential change in the loss they may cause and hence, do not successfully leverage the information from pretraining. However, at the same time, the general performance of all DST methods when starting with pretrained parameters on ImageNet is much worse in the low-density regime compared to their DST counterparts trained from scratch. This may suggest that the DST framework in the fine-tuning scenario is heavily biased by the pretrained weights and potentially does not efficiently explore possible connectivity patterns. These results suggest that one should be careful not to transfer the findings of the DST training to fine-tuning scenarios. In the second case, more research is needed to understand the behavior of the DST algorithms.

# I   Global versus Local Pruning

In our study, we have decided to focus on local pruning as this is the typical approach in the DST literature. Examples include RigL [12] (Algorithm 1), SNFS [11] (Figure 1, where pruning and redistribution occur layer-wise), SET [42]. In contrast, global pruning computes and compares the importance scores associated with the pruning criterion with all the parameters in the network. Hence, global pruning may change the sparsity distributions in each layer in an unsupervised manner. Note that global pruning may also have one small disadvantage: It may carry a higher risk of disconnecting the model if the importance scores $s(\theta_{i,j}^l)$ are not balanced across different layers. Using local pruning may avoid this risk and omit the need for investigating layer-wise normalization schemes.

However, we do consider the question of how the studied pruning criteria perform when paired with global pruning an interesting one. We perform a small experiment on ResNet-56 with CIFAR10 in which we compare the different pruning criteria in the global- and local-pruning setup on densities

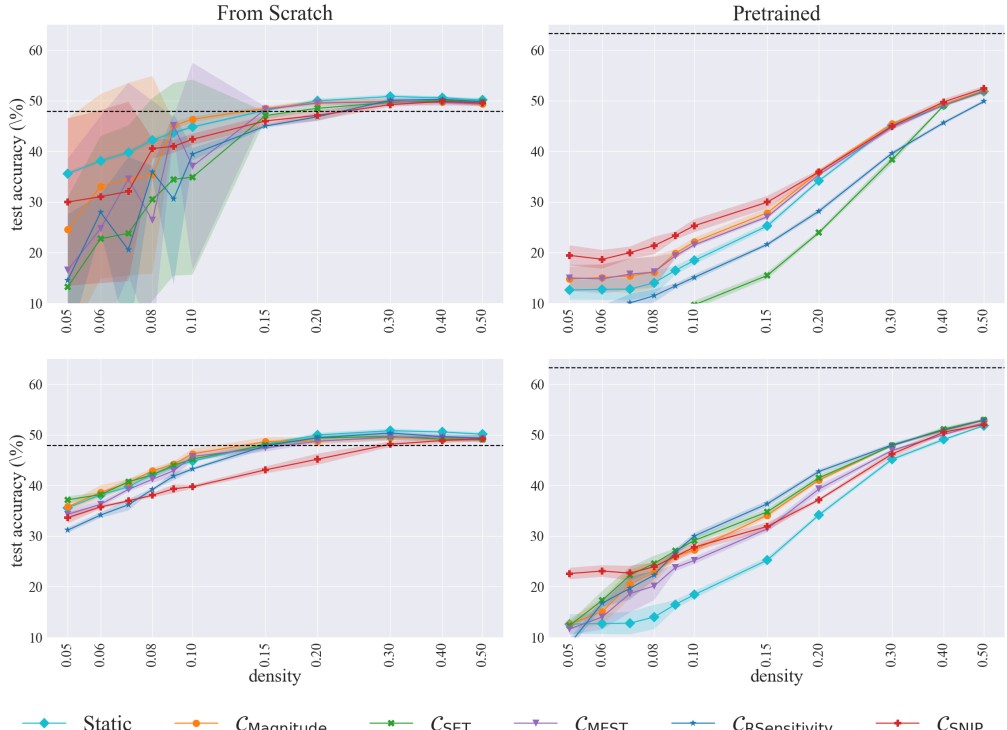

Figure 14: The performance of different pruning criteria on the EfficientNet architecture trained from scratch (left) and fine-tuned on the weights from training on ImageNet (right). The first row corresponds to random growth, and the second row corresponds to gradient growth. We can observe that in the setup that uses the pretrained weights (right column), the $\mathcal{C}_{\mathcal{SNIP}}$ criterion performs much better than in training from scratch. However, at the same time, the general performance of all DST methods is much worse in the low-density regime compared to the training from scratch.

0.05, 0.1, and 0.15. We present the results in Figure 15.[19] We observe that the results obtained in both global- and local- setups are indeed similar, with a slight preference for global pruning. By analyzing the plots, we can see that, in this case, the ranking results of the pruning criteria from the main text are indeed preserved. We consider the subject of using global pruning a very intriguing topic to investigate further in future research.

## J Effect of Batch Size and Pruning Schedule on Update Period

When considered together with the batch size and dataset size, the update period can be interpreted as the number of data samples the model sees (and trains on) before performing the mask update. In consequence, one may expect that with the increase of the batch size, the best choice of the update period will shift to lower values, keeping the total number of seen examples similar. To visualize this effect, we run a comparison on the MLP-CIFAR10 setup and observe just this behavior (see Figure 16(c) and Figure 16(d)).

Another potential confounding factor is the pruning rate schedule. In all our main experiments, we used the cosine schedule, which is known to perform well in the DST setup [12]. In this section, we additionally test this schedule on the MLP-CIFAR10 setup against two different schedulers: linear and constant. The linear schedule multiplies every 600 optimization steps the current decay rate by a constant factor (0.99). The constant schedule keeps the prune fraction fixed at 0.5 throughout the whole training. The results are presented in Figure 16(a) and Figure 16(b). We observe that the choice of the pruning factor schedule does not significantly influence the best value of the update period.

---

[19]We leave the $\mathcal{C}_{\mathrm{SET}}$ out due to its high similarity to the magnitude pruning.

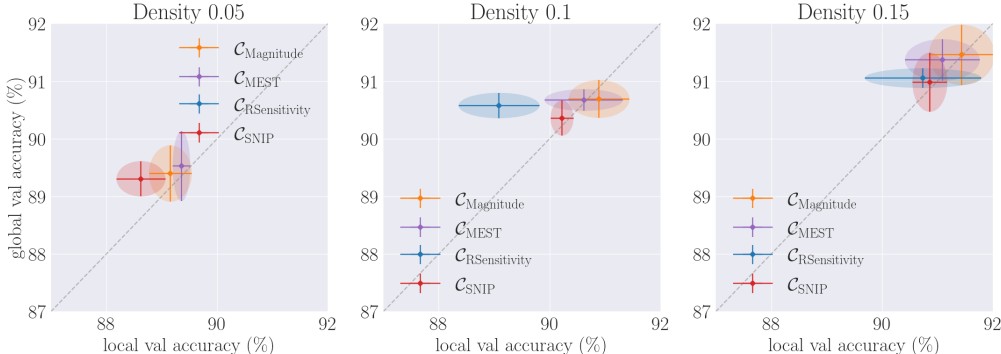

Figure 15: The test accuracy of ResNet-56 model using different criteria for densities 0.05, 0.1, 0.15. We plot the test accuracy obtained with global pruning (y-axis) versus the test accuracy obtained with local pruning (x-axis). Note that if the performance in both setups was the same, the values would perfectly lie at the gray, dashed diagonal line. Points above that line indicate that global pruning performed better.

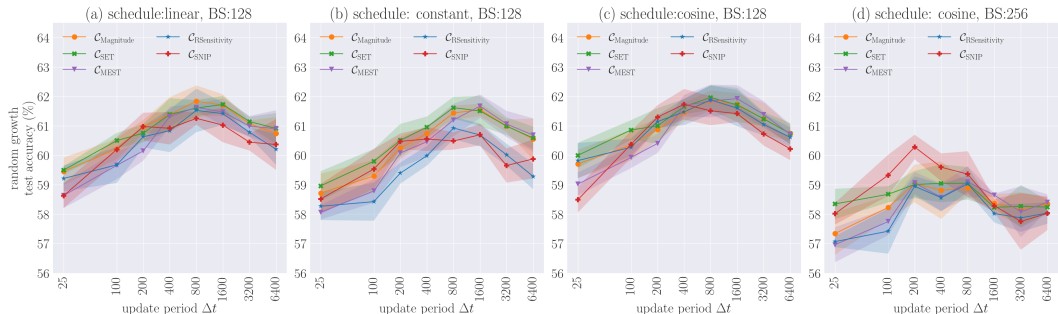

Figure 16: Additional results for the sensitivity to hyperparameters in the update-period study. We perform the study on random growth. From the left, we plot the results for **(a)** linear scheduler and default (i.e., 128) batch size (BS), **(b)** constant scheduler and default batch size, **(c)** cosine scheduler and default batch size (the same setup as in main paper),**(d)** cosine scheduler and batch size 256.

## K    Additional Jaccard Index Plots

We present the Jaccard index values computed after the first topology update of the model for different initial pruning ratios $\rho$ in Figure 17. Naturally, for smaller $\rho$, the difference becomes more visible. The $\mathcal{C}_{\mathrm{Magnitude}}$, $\mathcal{C}_{\mathrm{MEST}}$ and $\mathcal{C}_{\mathrm{SET}}$ criteria choose similar sets of weights to prune, while $\mathcal{C}_{\mathrm{SNIP}}$ and $\mathcal{C}_{\mathrm{RSensitivity}}$ are more diverse in their selections.

In addition, we also study how the similarity between the chosen sets for removal changes with time. To this end, we run each criterion with the DST framework until the end of training. Next, every 3200 iterations, we examine what weights for removal would be chosen by the remaining pruning criteria at this point in training. Note that such a relationship does not need to be symmetric. We report the results in Figure 18.

Moreover, we analyze the statistics of the never-removed connections. To be precise, we consider the masks at the end of training for the MLP, CNN, and ResNet-56 models on CIFAR10. Next, we measure the number of weights that are always retained by every pruning criterion for the same seed and we divide it by the number of all remaining (unpruned) weights in the model. We do the same on the negation of the masks to get the number of the always removed weights and divide it by the number of all removed weights of the model. We present the results in Figure 19. For the MLP model, around 23% of all the retained weights are shared across all the models. For CNN, this number is higher and goes up to 37%, while for the ResNet-56 it is equal to 11%. In general, the smaller overlap of the pruning methods for the ResNet model is also something that we observe in the main paper in Figure 6b.

Finally, we report the standard deviations for the Jaccard Index plots from the main text in Figures 20 and 21. Note that the mean Jaccard Index for the sets chosen for removal in the main text was computed by first taking a mean over different seeds for each layer and then averaging the results over all layers. Hence, in Figure 20, the standard deviation is calculated for each layer separately and then averaged. For the end-similarity, the Jaccard Index was computed by first taking the mean over all layers for a given seed and then taking the mean over all random seeds. Hence, in Figure 21, we report the standard deviation for averaging over the different random seeds.

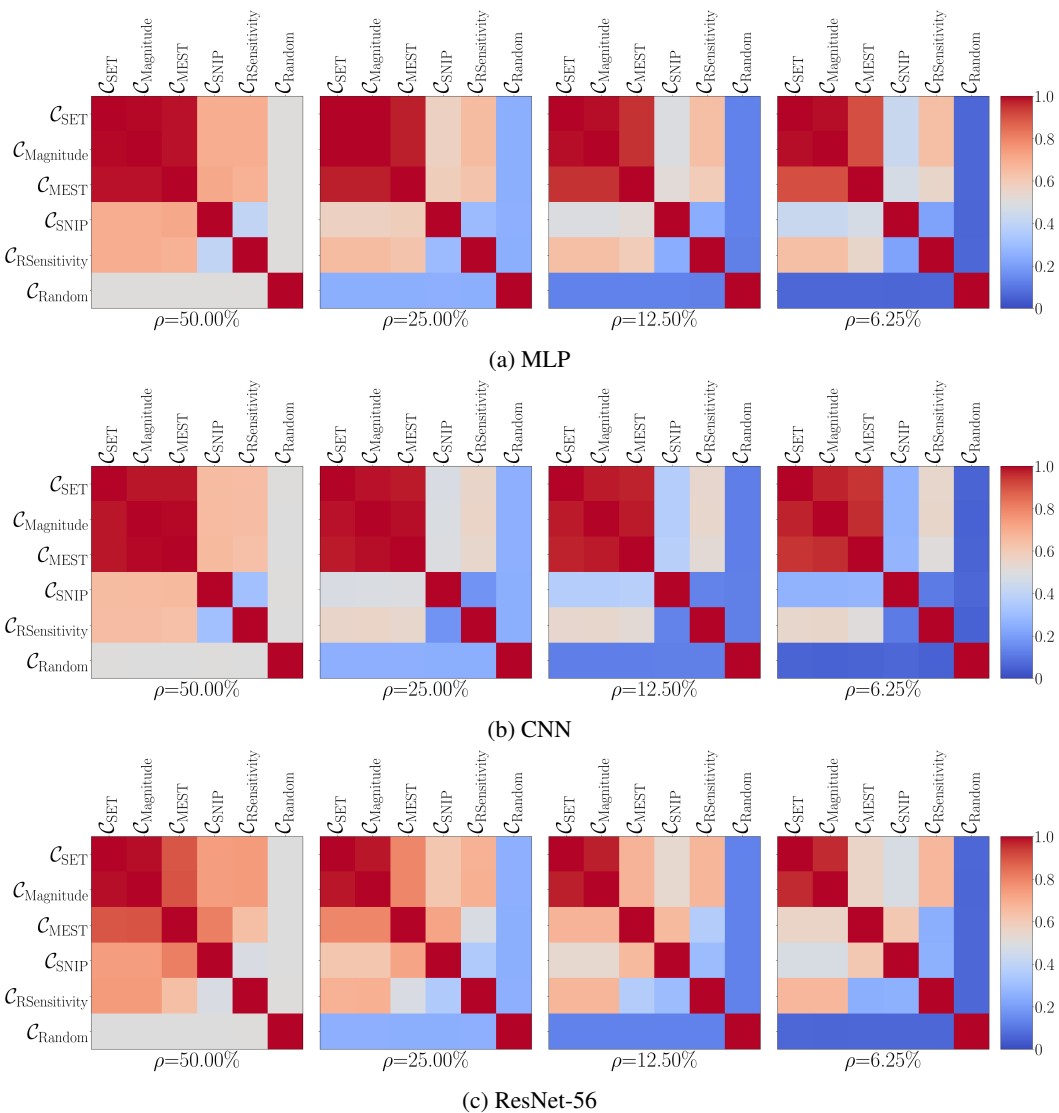

Figure 17: The Jaccard index between the selected pruning criteria computed for the same network state. The values are averaged over 5 different seeds. We observe that the $\mathcal{C}_{\text{Magnitude}}$, $\mathcal{C}_{\text{MEST}}$ and $\mathcal{C}_{\text{SET}}$ criteria select similar sets of weights to prune.

## L   Learning Curves

In this section, we plot the validation accuracy during the training for different densities and pruning criteria for the small and large MLPs and ConvNets. We notice that $\mathcal{C}_{\text{SNIP}}$ and $\mathcal{C}_{\text{RSensitivity}}$ criteria often lead to instabilities during the learning - see Figures 23, 24, 25, 26, 27, 28, 29, 30.

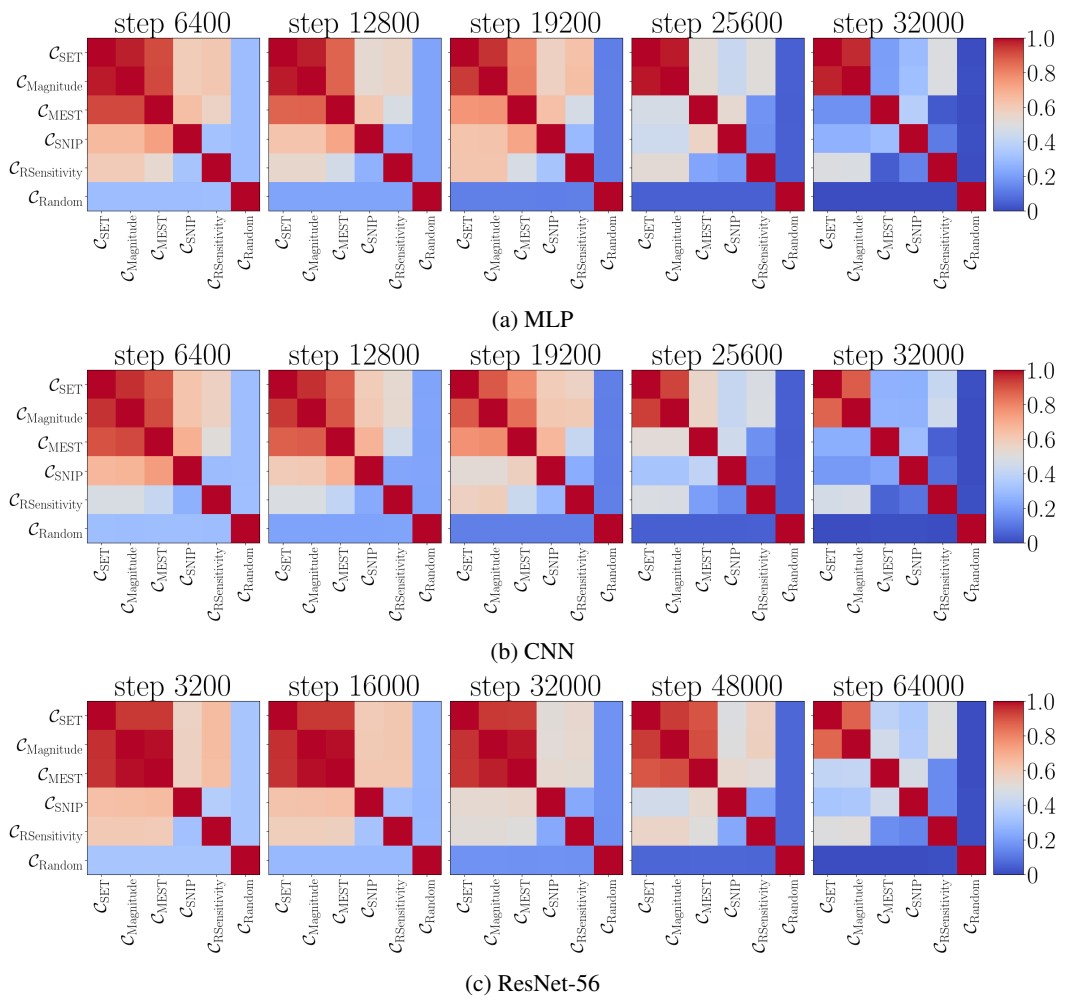

Figure 18: The Jaccard index between the selected pruning criteria in time. For each of the methods in the rows, we run the standard DST procedure. Then, every 3200 iterations, we verify what set of weights would be chosen at such a point by the remaining pruning criteria. Note that this relation does not need to be symmetrical.

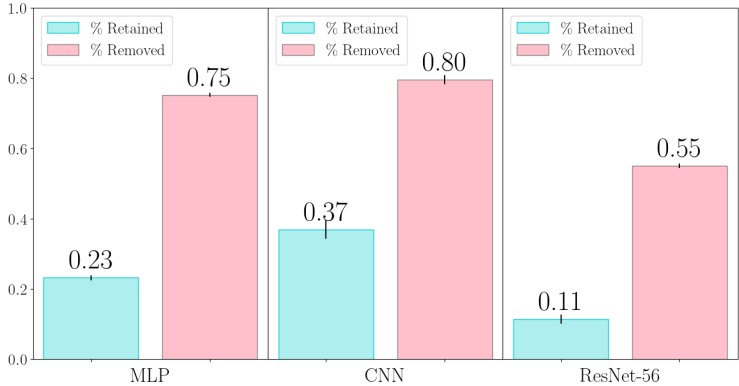

Figure 19: The number of weights retained by *all* the pruning criteria relative to the number of all unpruned weights (in blue). The number of weights removed by *all* the pruning criteria relative to the number of all pruned weights (in pink).

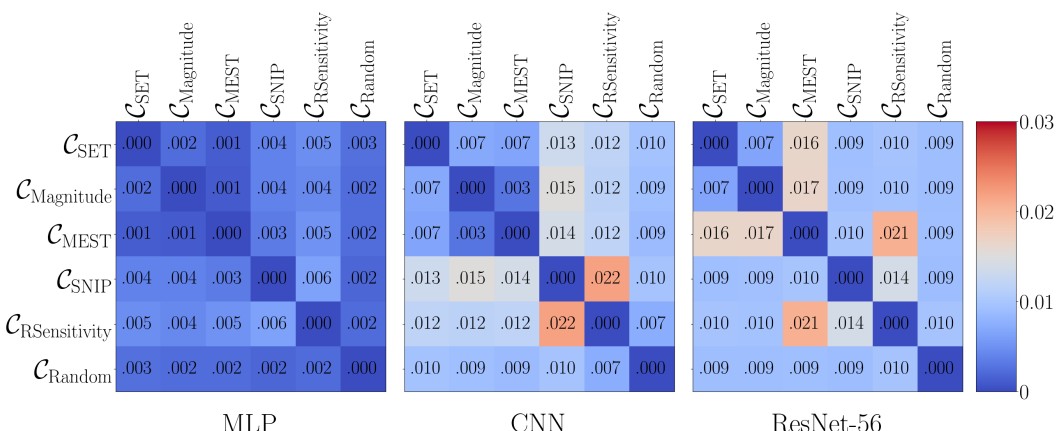

Figure 20: The mean standard deviation computed for the Jaccard index between the sets of weights chosen for removal during the first update of the sparse connectivity.

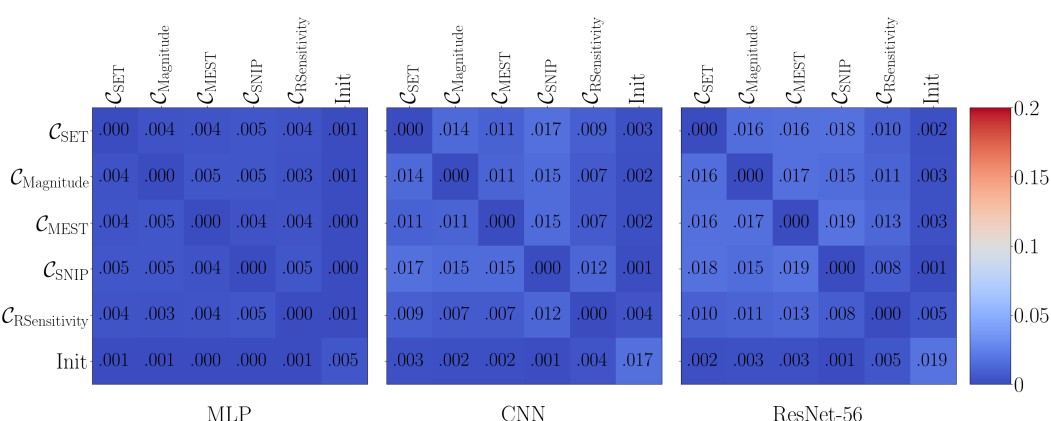

Figure 21: The standard deviation of the same index computed between the masks obtained by different pruning criteria at the end of training. The rows and columns represent the pruning criteria between which the index is computed.

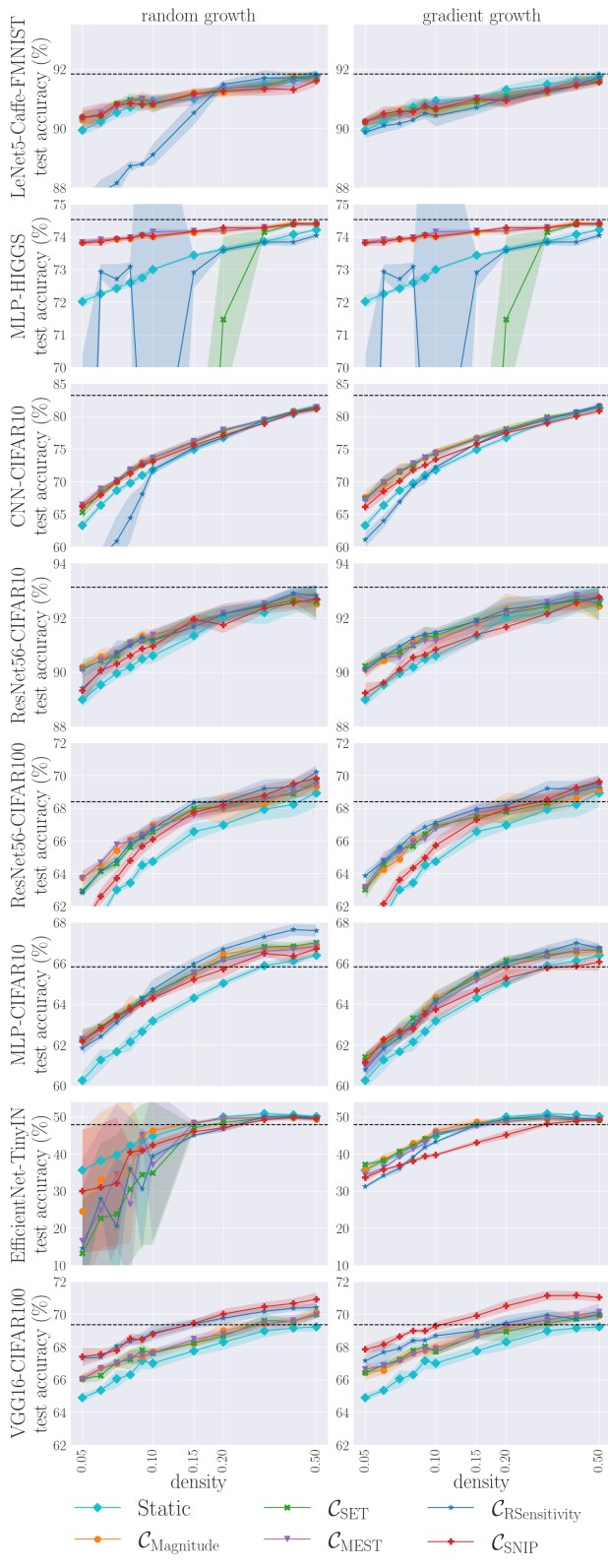

Figure 22: The performance of the pruning criteria with respect to the density on different datasets. Each row corresponds to a different model-dataset pair. The first column corresponds to the random growth, the second one corresponds to the gradient growth. Please note the logarithmic scale in the x-axis. We may observe that the differences between the methods become more visible predominantly in the low-density regime. DST almost always outperforms the static sparse training (cyan line).

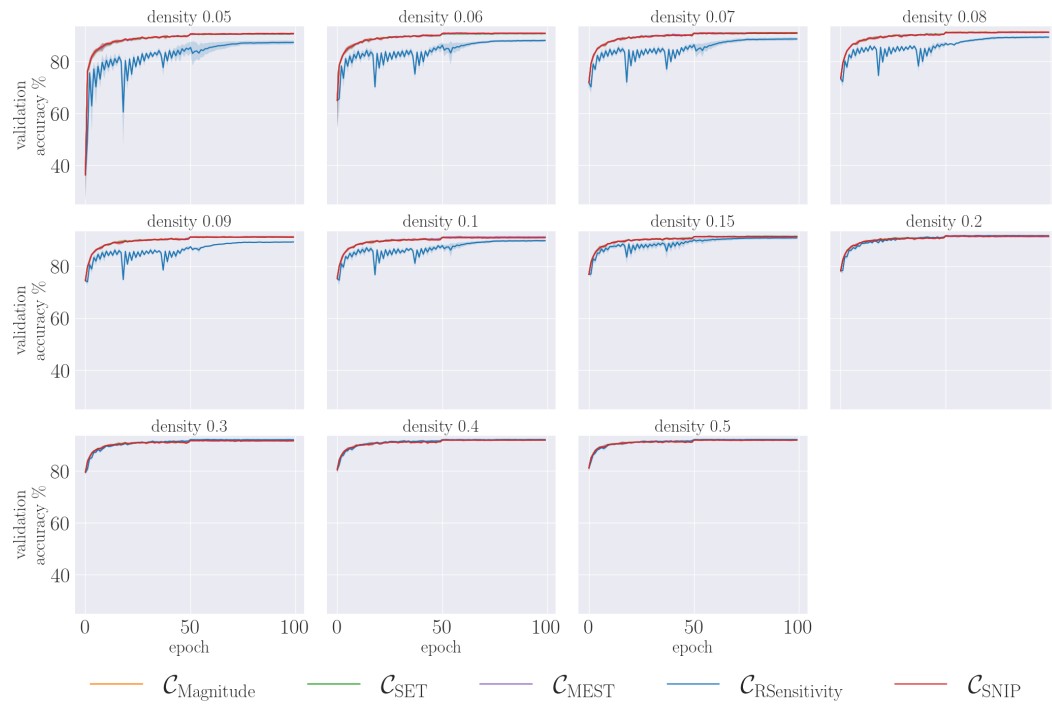

Figure 23: The validation accuracy versus the training epoch for different pruning criteria and densities on the LeNet-5-Caffe (FashionMNIST dataset).

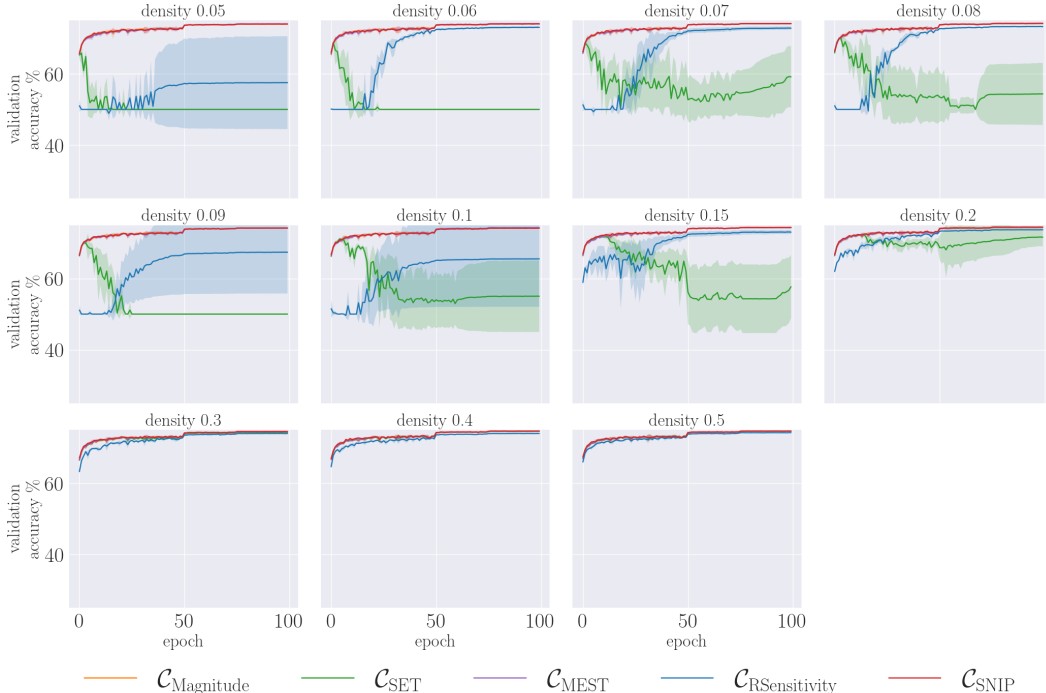

Figure 24: The validation accuracy versus the training epoch for different pruning criteria and densities on the small-MLP (Higgs dataset).

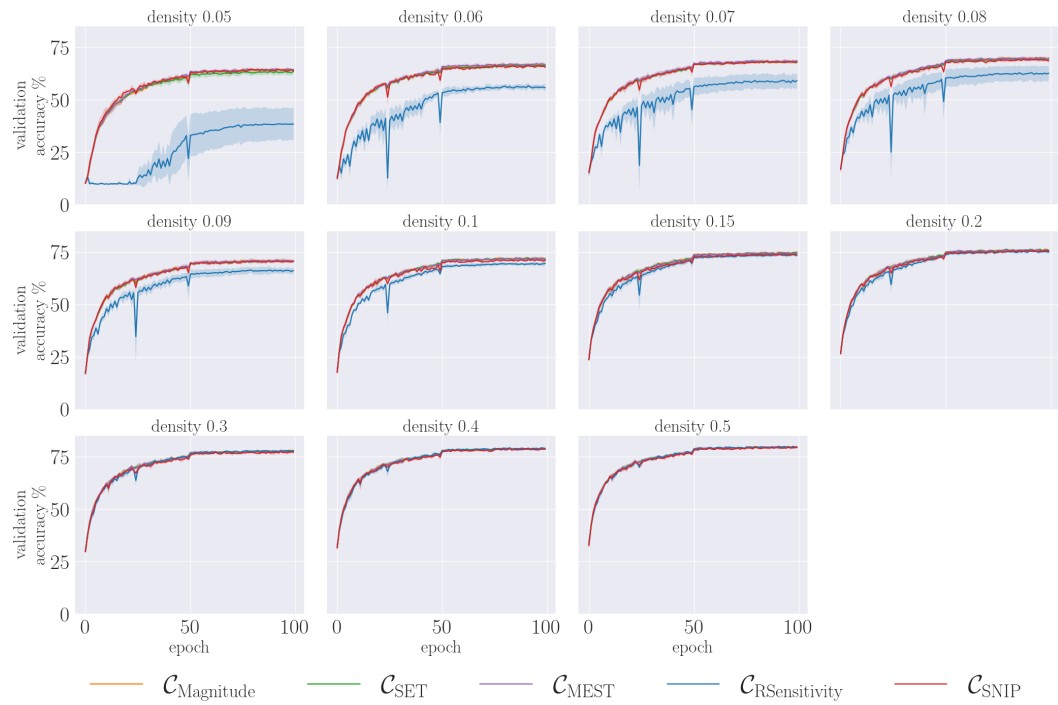

Figure 25: The validation accuracy versus the training epoch for different pruning criteria and densities on the small CNN (CIFAR10 dataset).

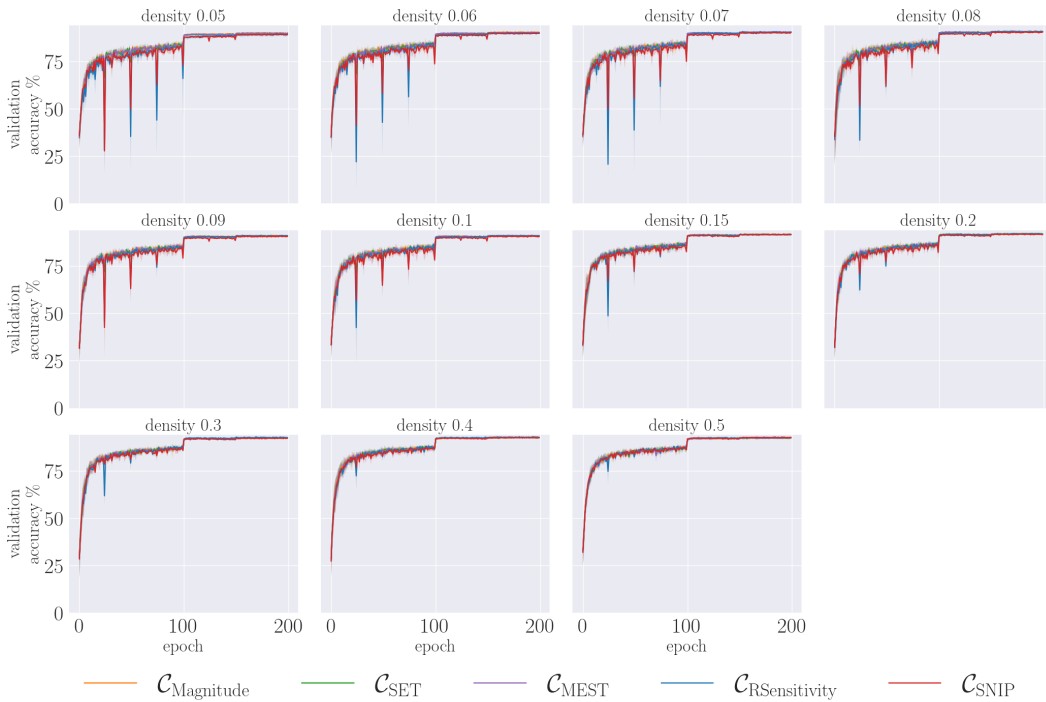

Figure 26: The validation accuracy versus the training epoch for different pruning criteria and densities on the ResNet-56 (CIFAR10 dataset).

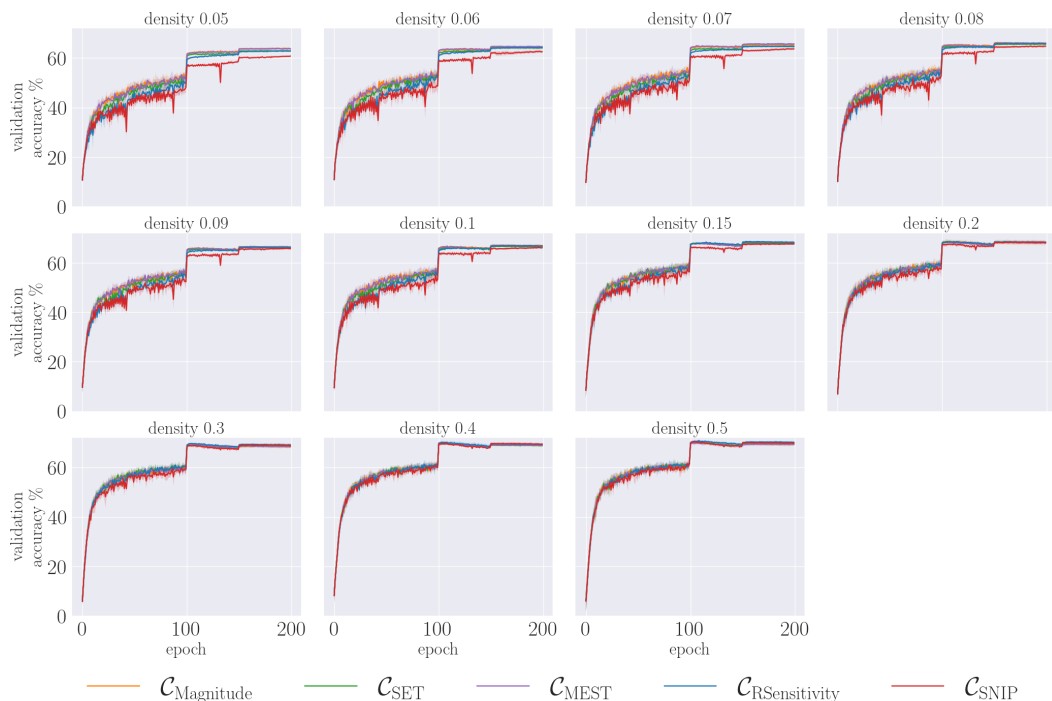

Figure 27: The validation accuracy versus the training epoch for different pruning criteria and densities on the ResNet-56 (CIFAR100 dataset).

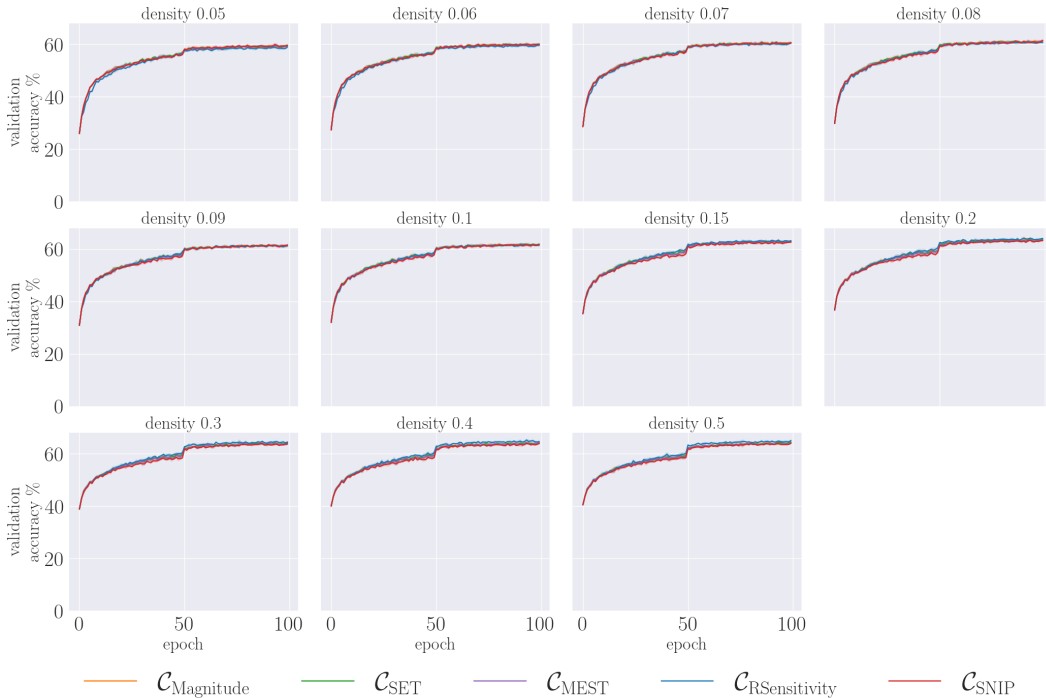

Figure 28: The validation accuracy versus the training epoch for different pruning criteria and densities on the large MLP (CIFAR10 dataset).

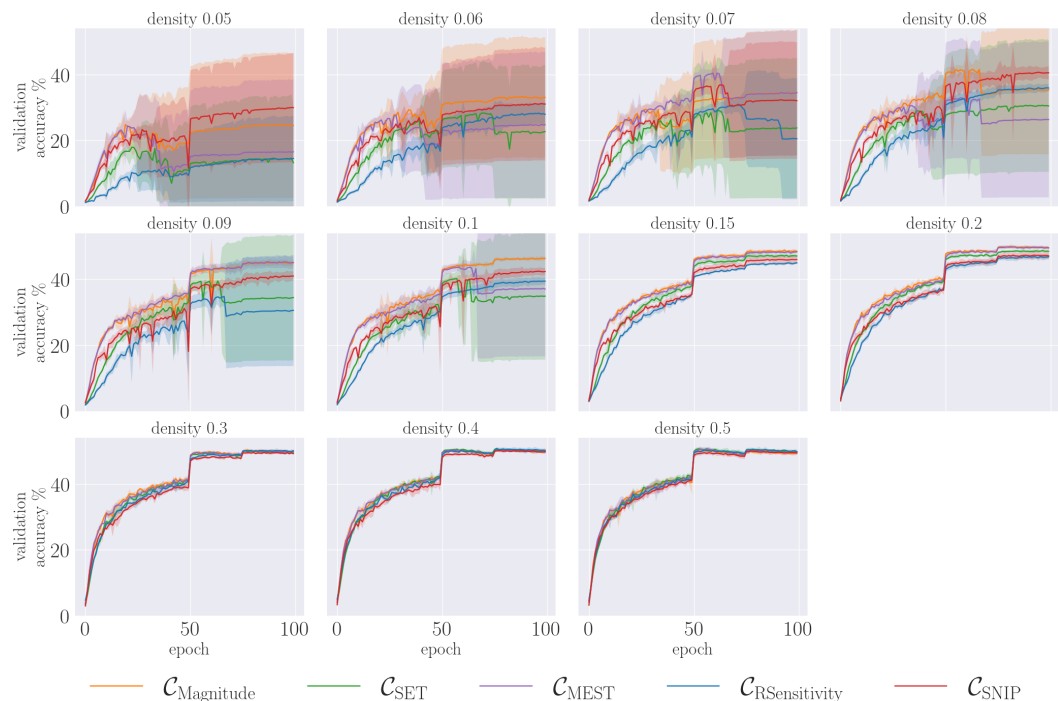

Figure 29: The validation accuracy versus the training epoch for different pruning criteria and densities on the EfficientNet (TinyImagenet dataset).

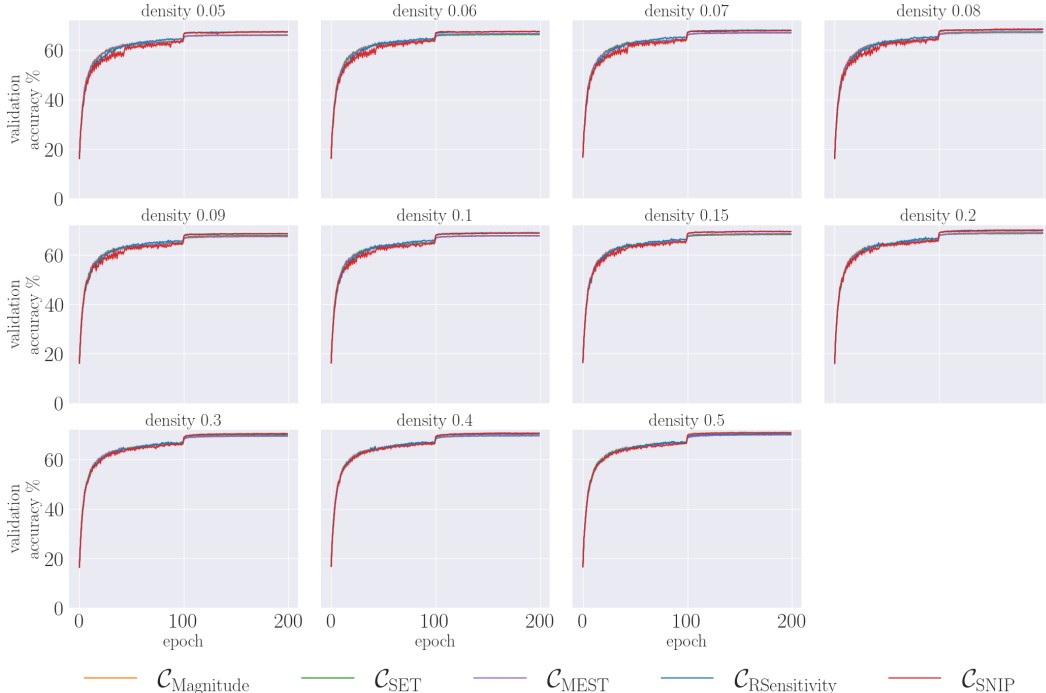

Figure 30: The validation accuracy versus the training epoch for different pruning criteria and densities on the VGG (CIFAR100 dataset).

