# OpenReview forum: "Fantastic Weights and How to Find Them: Where to Prune in Dynamic Sparse Training"
_NeurIPS.cc/2023/Conference — NeurIPS 2023 poster_

### Official Review · Reviewer_3Q28 · 2023-07-06

**Soundness:** 3 good
**Presentation:** 3 good
**Contribution:** 2 fair
**Rating:** 6
**Confidence:** 4

**Summary:**

The paper conducts an extensive empirical study including several popular pruning criteria and analyzes their impact on the DST framework on diverse models. They found that within a stable DST hyperparameter setup, the majority of the studied criteria
perform similarly, regardless of the model architecture and the selected growth criterion. In addition, in very sparse regimes, the simplest magnitude-based pruning methods surpass any more fancy choices.

**Strengths:**

1. The paper is well-written and the key research questions examined are well-supported by extensive experiments and appendix results.
2. Experiments related to how the frequency of topology updates influences the effectiveness of different pruning methods are interesting.


**Weaknesses:**

The primary concern related to the submission is the lack of novelty. Although some observations will be very beneficial to the sparsity community, the supremacy of magnitude-based weight removal in the high sparsity range is not novel. I believe extending this paper beyond small-scale models, to mid-to-large-scale models will make the paper more empirically strong and relevant in the current state of deep learning. In addition, some analysis of growth trajectory, connections which were never removed or always remained pruned across different pruning staretgies will add more value and uplift the quality of the work. Overall, it is still a strong empirical paper.

**Questions:**

The paper's readability will benefit by including important hyperparameters like prune and growth ratios etc in tabular format in the main draft. When written with experimental discussion, it becomes difficult to follow up.

**Limitations:**

See above.

---

> ### Author Rebuttal · Authors · 2023-08-09
>
> **Response Reviewer 3Q28**
>
> We are grateful to the Reviewer for the thoughtful comments and feedback. We are pleased to hear the reviewer considered our research questions well-examined by the extensive experiments we performed. We are also thrilled to see that the Reviewer found our experiments on the topology updates interesting and recognizes our work as a strong empirical paper. Below we address the raised concerns:
>
> *The primary concern related to the submission is the lack of novelty.* - We respectfully ask the reviewer to recognize the diverse possibilities for novelty within scientific research.  While introducing new algorithms to address problems is undoubtedly one facet of it, comprehending existing solutions holds equal significance. Our thorough and comprehensive study encompasses novelty by asking questions that have not been previously studied, such as to what extent the performance of DST is guided by the pruning criterion, and how the pruning criterion interacts with the choices of DST hyperparameters.
>
> We kindly encourage the Reviewer to consider that the significance of a weight (and thus the pruning criterion) can diverge from that in standard post-training pruning due to the parameter changes inherent in the DST framework (elaborated upon in lines 43-46 of the paper). Therefore it could be potentially dangerous to transfer the knowledge from standard (or static) pruning into DST without careful consideration. It is also worth noting that no definitive superiority of the magnitude criterion for DST has been previously firmly established, considering the emergence of various pruning criteria for DST in recent years. Our research offers valuable insights into the DST framework, suggesting that updates need not occur at such high frequencies and that more intricate pruning criteria do not necessarily outperform simpler magnitude-based ones. The originality and novelty of our work have been acknowledged by Reviewers xwEQ and mfmv. Therefore, we earnestly appeal to the Reviewer to assess our work within the broader context of innovation.
>
> *I believe extending this paper beyond small-scale models, to mid-to-large-scale models …* - We consider the VGG-16, ResNet-50, and Efficient-Net models as large enough to consider them at least “mid-scale” models (all models have above 10M parameters, with ResNet50 having 25M parameters). Both ResNets and Efficient-Nets are also commonly used in modern research, and we have chosen them in our evaluations because they have been included as the largest models in most of the works that used the studied pruning criteria (see Yuan et al. 2021, Evci et al. 2019). However, following the Reviewer’s request, we also include an evaluation on the ROBERTa-Large model (354M parameters) on the CommonsenseQA text dataset (please see the “Joint Response” for more details).
>
> *In addition, some analysis of growth trajectory ...* - Thank you for this suggestion. In order to improve our work, we provide additional statistics on the never removed connections. We consider the masks at the end of training for the MLP, CNN, and ResNet-56 models on CIFAR10. Then we measure the number of weights that are always retained by every pruning criterion for the same seed and we divide it by the number of all remaining (unpruned) weights in the model. We do the same on the negation of the masks to get the number of the always removed weights and divide it by the number of all removed weights of the model. We present the results in the Rebuttal PDF in Figure R4. For the MLP model around 23% of all the retained weights are shared across all the models. For CNN, this number is higher and goes up to 37%, while for the ResNet-56 it is equal to 11%. In general, the smaller overlap of the pruning methods for the ResNet model is also something that we observe in the main paper in Figure 6b in the main text. Additionally, we would like to note that in Figure 9 in Appendix I we report how the Jaccard Index between the sets of weights selected for removal changes during the training - perhaps the Reviewer will also find this plot interesting.
>
> **Questions:**
>
> The paper's readability will benefit by … - Thank you, we will add the below table to the Appendix:
>
> | Experiment | Prune Fraction  | Prune Fraction Decay Scheduler | Density                                                   	| Update Period                          	|
> |------------|-----------------|--------------------------------|---------------------------------------------------------------|--------------------------------------------|
> | Fig. 1   | 0.5         	| cosine                     	| 0.05, 0.06, 0.07, 0.08,  0.09, 0.1, 0.15, 0.2,  0.3, 0.4, 0.5 | 800                                    	|
> | Fig. 2   | 0.5         	| cosine                     	| 0.2                                                       	| 800                                    	|
> | Fig. 4   | 0.5         	| cosine                     	| 0.2, 0.1                                                  	| 25, 50, 100, 200, 400, 800, 1600,3200,6400 |
> | Fig. 5   | 0.5         	| cosine                     	| 0.5                                                       	| 800                                    	|
> | Fig. 6a  | 0.5         	| cosine                     	| 1.0                                                       	| 800 (measured  only once)              	|
> | Fig. 6b  | 0.5         	| cosine                     	| 0.2                                                       	| 800                                    	|
>
> Once again we grateful for the feedback provided by the Reviewer and we hope that our responses have addressed the raised concerns. We kindly ask the Reviewer to reconsider adjusting the score accordingly.
>
> References:
> Evci, Utku, et al. "Rigging the lottery: Making all tickets winners." ICML. PMLR, 2020.
> Yuan, Geng, et al. "Mest: Accurate and fast memory-economic sparse training framework on the edge." NeurIPS 34 (2021).

---

> > ### Comment · Reviewer_3Q28 · 2023-08-16
> > **Rebuttal Response**
> >
> > I have read the author's rebuttal. Some of my concerns are addressed. I update my score to 6.

---

> > > ### Author Response · Authors · 2023-08-16
> > >
> > > We thank the Reviewer for reading our rebuttal, and we are grateful for updating the score. We are happy to provide any additional clarifications if needed.

---

### Official Review · Reviewer_mfmv · 2023-07-06

**Soundness:** 3 good
**Presentation:** 3 good
**Contribution:** 3 good
**Rating:** 7
**Confidence:** 4

**Summary:**

This paper performs a systematic study of pruning strategies for dynamic sparse training (DST) methods, comparing their performance and structural decisions across backbone architectures, datasets, structural change frequencies, connection densities, and batch sizes.

**Strengths:**

Originality: This work is the first systematic study on DST methods, combining various pruning and growing strategies that have thus far only been studied in isolation. It is also the first work to structurally compare the results of various methods. It does not introduce any new DST strategies, although it adapts a well-known pruning metric (SNIP) for the DST context and makes novel combinations of known DST strategies.

Quality: This paper backs up claims with adequate experimentation and analysis, although the scope is limited to mostly image classification tasks (plus one tabular dataset).

Clarity: The paper is easy to understand overall, with a clear writing style.

Significance: This paper provides a significant study on the interplay not only of growing and pruning strategies but also factors such as backbone architecture, dataset, structural change frequency, connection density, and batch size.


**Weaknesses:**

I think random pruning (paired both with random or informed gradient-based growth) would be an interesting baseline in addition to dense and statically sparse architectures.

As noted by the authors, expanding to tasks beyond image classification (and one tabular dataset) would broaden the scope of this work.

You claim that growing is more thoroughly studied than pruning, but I’m not sure this is true. In Line 40, Dettmers and Zettlemoyer (2019) is cited as emphasizing the growth criterion but rather, to my understanding, uses random growth with informed pruning. There seems to be a roughly equal number of cited DST works that use simple pruning with more informed growing to those using simple growing with more informed pruning. This paper still has significance by focusing on the pruning method (although more growing methods could be included, but this would quickly increase the experimentation cost), but I think this language should be clarified.

The analysis on update period in Section 5.3 is interesting, but I think should be qualified by noting how the pruning factor schedule and batch size may be confounding factors.


**Questions:**


In addition to the batch size studies, I would be interested to see (in this or future work) how much the batch used for gradient evaluation affects the connections selected for pruning and/or growth in gradient-based pruning/growth methods. This could be studied by performing the training as normal until immediately before the 1st pruning step, then repeatedly resampling a batch and performing a backwards pass (without updating weights) in order to record which connections would be pruned and which would be grown.

Is Figure 2 aggregated over multiple trials per protocol? If so, please add error bars. If not, please run multiple trials.

Is there any way to add error intervals to Figure 6? I like the graphical presentation but would also appreciate error intervals to determine how noisy the metrics are and how significant the differences between strategies are.

Typos and tips (not considered negatively against paper):
* Line 13: “cautions”?
* Footnotes referring to appendices are not necessary: placing the references directly in the main text may save space.
* The paper layout and figure placement was a bit hard to follow. Figures that span across the whole width seem to be placed rather randomly, while they are conventionally placed at the top of a page. This helps the reader read the main text more continuously and find the appropriate figure when referenced.
* Many captions ended with an analysis of the figure, like the last two sentences of the caption of Figure 1: I would prefer to see this in the main text rather than the caption.
* The Jaccard index equation could be moved to an appendix to save space.


**Limitations:**

Limitations and broader impacts are explicitly addressed.

---

> ### Author Rebuttal · Authors · 2023-08-09
>
> **Response to Reviewer mfmv**
>
> We thank the Reviewer for the review. We are very happy to see that the Reviewer appreciates the originality and significance of our work, recognizing it as the first systematic study on pruning criteria in DST methods. Furthermore, we are glad to hear our experimental evaluation and analysis have been acknowledged as adequate. We were mostly thrilled to receive such positive feedback. In the following, we address the remaining questions raised by the Reviewer:
>
> *I think random pruning (paired both with random or informed gradient-based growth) would be an interesting baseline…* - We have observed that random pruning paired either with random or gradient growth performs much worse than all of the other criteria and baselines (we have added a visualization of that effect on ResNet56 on CIFAR-10 in Figure R3 in the rebuttal PDF). The performance drops especially dramatically for the very low densities, making it a less interesting baseline. While random may be a quite reasonable choice for the growth (due to influencing the exploration), in pruning it effectively leads to removing arbitrary connections in a trained network.   For the above reasons, we initially decided to use only dense and static sparse baselines. We will add the random performance for all the Figures for the camera-ready version.
>
> *As noted by the authors, expanding to tasks beyond image classification (and one tabular dataset) would broaden the scope of this work…* - We have enhanced our empirical evaluation with results on the ROBERTa Large model (~354M parameters) on the CommonsenseQA text dataset. We kindly ask the Reviewer to refer to the “Joint Response” for details.
>
> *You claim that growing is more thoroughly studied than pruning, but I’m not sure this is true…* - Thank you for this comment. Indeed, that phrase can take benefit of some clarifications. What we intended to convey is that pruning criteria have not ever been rigorously compared in DST. At the same time, the interplay between some growth criteria is rather better understood (as demonstrated, for example, by Evci et al., 2019). Additionally, we would like to kindly note that we believe that Dettmers and Zettlemoyer (2019) do, in fact, employ magnitude pruning alongside momentum-based growth (as evidenced by Algorithm 1 in their paper). We think rephrasing lines 38-41 from our paper to "Much of the research concerning design choices in DST focuses on the analysis of methods where variations occur in the growth criterion or the pruning criterion becomes intertwined with other design considerations [3, 10, 9, 1] (...)." will clarify our point. We hope that this adjustment effectively addresses the concerns raised by the Reviewer.
>
> *The analysis on update period in Section 5.3 is interesting, but I think should be qualified by noting how the pruning factor schedule and batch size may be confounding factors.* - Thank you for raising this point. The update period, when considered together with the batch size, can be interpreted as the number of data samples the model sees (and trains on) before performing the mask update. We will adjust the text in the Camera Ready version of the paper to make sure it includes that information. In consequence, we would expect that once we increase the batch size, the best choice of the update period shifts to lower values, keeping the total number of seen examples similar. We have run a quick comparison on the MLP-CIFAR10 setup and observed this behavior (see Figure R1c and R1d in the rebuttal PDF). Regarding the pruning factor schedule: we again run a quick comparison on the MLP-CIFAR10 setup, and try two different schedulers, apart from the default cosine one: the linear one, and the constant one. The linear schedule multiplies every 600 optimization steps the current decay rate by a constant factor (0.99). The constant schedule simply keeps the prune fraction fixed at 0.5 throughout the whole training. The results are in Figures R1a, R1b, and R1c in the Rebuttal PDF. We observe that the choice of the pruning factor schedule does not significantly influence the best value of the update period. We will add those results to the Appendix.
>
> **Questions:**
>
> *I would be interested to see (in this or future work) how much the batch used for gradient evaluation affects the connections selected for pruning and/or growth in gradient-based pruning/growth methods...* - We have run for the rebuttal a quick experiment where we only increase the batch size for the DST update (from 128 to 1024). The results are in Figure R5 in the Rebuttal PDF. We observe that in general, it does not enhance the performance (and works significantly worse with gradient growth than with random growth for the MLP-CIFAR10). We consider the study on the interplay between DST update batch size and the model performance as an interesting direction for future work.
>
> *Is Figure 2 aggregated over multiple trials per protocol?* - Yes, we have run 3 experiments in each setting, and we did include the error bars in the Figure  - they are just very small (all the methods perform very stable). We apologize, we will enlarge them in the Figure to make them more visible.
>
> *Is there any way to add error intervals to Figure 6?* - Thank you, this is a valid point. We will add the error intervals to the Appendix.
> Typos and tips (not considered negatively against the paper): Thank you for catching the typos and providing the tips. We will enhance our paper accordingly.
>
> We once again appreciate the Reviewer’s favorable rating and constructive feedback, helping us to further enhance the quality of the paper.

---

> > ### Comment · Reviewer_mfmv · 2023-08-14
> > **Response to rebuttal**
> >
> > Thank you for your response.
> >
> > Regarding my question on how much the batch used for gradient evaluation affects the connections selected for pruning and/or growth in gradient-based pruning/growth methods, I meant the batch and not the batch size. Specifically, how sensitive is each method to the samples used for measurement? Your referenced experiment only partially studies this question, as we can assume a larger evaluation batch may give a more stable measurement, but only comparing two batch sizes isn't quite thorough enough to make any conclusions on this topic.
> >
> > I maintain my initial rating.

---

> > > ### Author Response · Authors · 2023-08-16
> > >
> > > We appreciate the Reviewer's response and we apologize for misunderstanding the comment regarding the batch experiment. Exploring the influence of batch samples on the mask, beyond just batch size, is an intriguing question. We view it as an exciting avenue for future research. In response to the Reviewer's suggestion, we conducted a preliminary experiment using the ResNet56-CIFAR10 setup:
> > >
> > > We perform the training normally until immediately before the 1st pruning step. Next, we select a batch and conduct the forward and backward passes without updating the weights. We then perform the mask update and save the masks right after the pruning and right after the growth. We undo the update and repeat this procedure for every 10 batches. Then, for each pair of saved batches, we compute the mean Jaccard index of pruned weight sets and calculate the average over all pairs (“pruned mean" in the table below). Additionally, we measure the maximum and minimum Jaccard index obtained by comparing any weights in a layer between any two saved batches (referred to as “pruned max” and “pruned min”). We also calculate the mean of pairwise Jaccard indices after the **complete** mask update (i.e. pruning and gradient growth, referred to as “mask mean”). We present the results in the table below.
> > >
> > > | criterion |   pruned mean  |   pruned max |   pruned min |   mask mean |
> > > |:--|:--:|:--:|:--:|:--:|
> > > | $\mathcal{C}_{magnitude}$ | 		  1.000$\pm$0.000		 |    	  1.000  	  |    	  1.000  	  | 		  0.631$\pm$ 0.004 |
> > > | $\mathcal{C}_{SET}$|    	  1.000$\pm$0.000 	  |    	  1.000  	  |    	  1.000  	  | 		  0.631$\pm$ 0.004 |
> > > | $\mathcal{C}_{MEST}$| 		  0.989 $\pm$ 	0.001 |    	  1.000  	  |    	  0.934 | 		  0.629$\pm$  0.004 |
> > > | $\mathcal{C}_{RSensitivity}$ |   		0.592 $\pm$ 0.003 |    	  0.878 |    	  0.407| 		  0.562$\pm$0.003 |
> > > | $\mathcal{C}_{SNIP}$| 		  0.586 $\pm$      0.003  |    	  0.838 |    	  0.464 | 		  0.517$\pm$0.004 |
> > >
> > > As expected, the magnitude and SET criteria consistently prune the same sets across batches due to their reliance on global statistics, unaffected by batch changes. The RSensitivity and SNIP criteria exhibit less overlap in pruned sets between batches (around 0.592 and 0.586 respectively). However, the size of this overlap is relatively stable, as reflected by the rather small standard deviation. We consider extending this preliminary study an interesting direction for future work. We thank the Reviewer for raising our attention to this matter and for the provided positive feedback.

---

### Official Review · Reviewer_xwEQ · 2023-07-07

**Soundness:** 3 good
**Presentation:** 2 fair
**Contribution:** 3 good
**Rating:** 6
**Confidence:** 3

**Summary:**

This paper does a large scale study of dynamic sparse training, primarily on vision datasets. For a variety of models and hyperparameter settings, pruning using magnitude is found to be as or more effective when compared to pruning using more complex criteria proposed in the literature. The performance findings are reinforced by a study of the weights different pruning criteria select, and the impact of topology update frequency is studied.

**Strengths:**

Originality:

1. This paper provides the most comprehensive study of DST pruning criteria that I am aware of, providing a valuable insight about the relevance of the magnitude criterion.

Quality:

2. The experiments are designed well and support the contention that magnitude is a strong criterion in a variety of DST scenarios.

Clarity:

3. The submission is well written.

Significance:

4. The performance comparisons of criteria that have been shown to be effective in the literature (e.g. MEST and magnitude) are valuable for the community.


**Weaknesses:**

This paper does not have any major weaknesses. I believe it's significance to the community (and correspondingly my score) would be best enhanced by ensuring the experiments cover a broad range of relevant architectures. I address this and some more minor issues, along with my suggestions, below.

**Questions:**

1. In your related work, it is probably worth mentioning that Frankle et al. (2021) did an analysis of similar pruning criteria for static sparse training, also finding that magnitude was as good as more complex criteria.

2. Line 266: An alternative explanation might be that removing a smaller amount of weights leads to less disruption of the batch norm statistics with non-magnitude criteria. After pruning, do you update the batch normalization statistics (e.g., by running the model on the training data without updating its weights, see page 16 of [Zimmer et al., 2023](https://arxiv.org/pdf/2306.16788.pdf))? I ask because removing small magnitude weights may have little effect on these statistics, but removing larger weights (e.g., because their gradients were small) could have more substantial effects on the relevance of the old batch norm statistics to the new model. In which case, it would be difficult to conclude that the criterion is responsible for a certain performance level as that performance level could be attributable (at least in part) to incorrect batch statistics. It might be worth running an experiment to check this (e.g., on EfficientNet).

3. The networks considered cover a broad range but leave out architectures that have become more relevant in recent years. Also, the paper claims to cover large scale convolutional networks, but the largest network is a ResNet-50. Consider running on larger models and different architecture styles, e.g. ViTs.

4. Figures are too far from where they are discussed in the text. Consider adjusting their placement.

5. Lines 163-169 might benefit from being rephrased. As written, they do not provide a clear intuition for the expectation that a gradient based method would do better, which is what I think you are offering in these lines. In contrast, lines 184-185 (in conjunction with lines 177-178) give clear intuition for why I might expect a gradient based method to perform better.

6. Lines 188-189: wouldn't the mask solutions have to be "genuinely different" if the performances associated with the different criteria differed? I would say the prior experiments "do not necessarily indicate" instead of "do not indicate".

7. Line 270 and Figure 5 clash. You say that pruning adds regularization, but Figure 5 shows the dense model has the largest training loss - validation loss. Maybe you meant to define the loss gap as val - training instead of training - val.

**Limitations:**

The identified limitations are appropriate.

---

> ### Author Rebuttal · Authors · 2023-08-09
>
> **Response to Reviewer xwEQ**
>
> We appreciate the feedback from the Reviewer and the positive comments about both the originality and the value of our work. We are happy to see that the Reviewer finds our experiments well-designed and supportive of our main claims. We were also glad to learn that the Reviewer does not see any major weaknesses in our work. We address the remaining questions of the Reviewer below:
>
> 1. *In your related work, it is probably worth mentioning that Frankle et al. (2021)*  - For clarity, did the Reviewer mean the Pruning neural networks at initialization: Why are we missing the mark?." arXiv:2009.08576, ICLR (2021) paper? If so, it is indeed an important and related study, that contrary to our setup focuses on the **static** sparse pruning. Thank you for raising our attention to it, we will update the Related Work section accordingly.
>
> 2. *Line 266: An alternative explanation might be that…* - Thank you for raising our attention to this work. We agree that the bach normalization statistics may influence the pruning during evaluation. Please note, however, that in DST we always prune during training, where the batch normalization statistics are not used. During training, the normalization is computed on the statistics directly from the mini-batch (see Algorithm 1 of Ioffe & Szegedy 2015, and the documentation of TensorFlow and PyTorch - links in the references below). The running normalization statistics only become relevant when we compute the final evaluation accuracy on the test set. But before that happens, the model still uses the iterations between the last mask update and the end of the training to update the running statistics. Therefore at the time of evaluation, they should already be adjusted according to the new mask structure.
>
> 3. *The networks considered cover a broad range but…* - We have added a task on a transformer architecture (ROBERTa Large, ~354M Parameters) on the CommonsensQA dataset. Please refer to the “Joint Response” for details. There we also explain why we initially focused on convolutional and MLP-styled architectures.
>
> 4. *Figures are too far from where they are discussed…* - Thank you, we will fix that.
>
> 5. *Lines 163-169 might benefit from being rephrased…* - Thank you for noticing that. We will rephrase it to: “Since the gradient may provide information on how the weight will change in the future, it indicates the trend the weight might follow in the next optimization updates. Therefore it may be more suitable in the DST approach, where connectivity is constantly evolving and hence the insights into future updates may be beneficial”. We hope this clarifies the statement.
>
> 6. *Lines 188-189: wouldn't the mask solutions have to be "genuinely different" if the performances associated with the different criteria differed? …* -  Not necessarily. Performance (i.e., accuracy) is only a measure of the end result of the model. A large difference in performance does not have to imply (a priori) a large difference in the mask (e.g., hypothetically, the mask could differ in only a few connections, but those connections could be very significant to the performance). Without studying to what extent the masks of different pruning criteria differ (but considering only performance), we are not able to identify such a situation.
>
> 7. *Line 270 and Figure 5 clash …*  - Thank you for catching that. We had a typo in the text. Yes, Figure 5 shows the validation loss - training loss.
>
> We have to the best of our effort, addressed all the concerns raised by the Reviewer and followed the suggestions to include more contemporary and large architectures by adding the ROBERTa Large model. We hope that our answers provide the necessary clarifications needed to enhance our paper. We would greatly appreciate if the Reviewer could re-evaluate our work taking into account our response and the fact that the Reviewer finds that there are no major weaknesses in our paper.
>
> References:
>
> Frankle, Jonathan, et al. "Pruning neural networks at initialization: Why are we missing the mark?." arXiv:2009.08576, ICLR (2021).
> Ioffe, Sergey, and Christian Szegedy. "Batch normalization: Accelerating deep network training by reducing internal covariate shift." International conference on machine learning. pmlr, 2015.
> tensorflow batch norm: https://www.tensorflow.org/api_docs/python/tf/keras/layers/BatchNormalization
> pytorch batch norm: https://pytorch.org/docs/stable/generated/torch.nn.BatchNorm2d.html.

---

> > ### Comment · Reviewer_xwEQ · 2023-08-15
> > **Acknowledgement of rebuttals and reviews**
> >
> > I have read the reviews and rebuttals, and I have updated my score. This submission's analysis of dynamic pruning criteria in various contexts is valuable, and my improved score reflects the addition of the ROBERTa analysis and the addressing of my more minor comments.
> >
> > Comments on your responses to my questions:
> > 1. Yes, that paper. Sounds good.
> > 2. Your response suggests there are enough iterations between the last mask update and the end of training for this to not be a concern, thanks!
> > 3. Sounds good.
> > 4. Sounds good.
> > 5. Sounds good.
> > 6. I agree that a large difference in performance does not imply a "large" difference in the mask, but I think a large (or a small) difference in performance does imply a "genuine" difference in the mask (at least at some point during training, assuming every training facet but the masking is held constant). If you agree, then you could address this concern by saying "significantly different" instead of "genuinely different".
> > 7. Sounds good.

---

> > > ### Author Response · Authors · 2023-08-16
> > >
> > > We express our gratitude to the Reviewer for thoroughly reviewing our response. Concerning point 6, we agree that substituting "genuinely" with "significantly" can enhance precision and we will adjust the text accordingly. We are happy that our answers addressed the Reviewer’s concerns, and we thank the Reviewer for raising the score.

---

### Official Review · Reviewer_aJDG · 2023-07-09

**Soundness:** 3 good
**Presentation:** 4 excellent
**Contribution:** 2 fair
**Rating:** 5
**Confidence:** 4

**Summary:**

This empirical paper looks into the setting of adapting neural network architecture during training (dynamic sparse training), repeatedly pruning and growing the network. Authors try to draw conclusions about the performance and topology of various pruning criteria. They show that in high density regimes, most of criteria give similar results, with methods that incorporate gradients performing best, but in low density regime, simple magnitude based pruning performs the best

**Strengths:**

- well written, easy to follow
- author tackle the question that has not been investigated before
- the emprical study is rigorous/well executed (modulo my questions/remarks)
-assuming authors opensource their benchmark, it can serve as a test bed for new pruning and /or growth criteria


**Weaknesses:**

- Limited novelty (empirical study of existing methods). Not very deep insights
- mostly vision data and only 1 tabular dataset. No text data


**Questions:**

This is comprehensive and well executed and I am willing to raise my score if the authors have good intuition for the questions below.

1) My conceptual question is related to line 218 - you are using the same hyper parameters for all pruning criteria. I wonder whether some pruning methods that are more crude for example can benefit from smaller learning rate, etc - basically I am worried that the reuse of all the same hyper parameters is what gives you the results that are all similar to each other. Because in reality, I assume when you train with DST and you set the pruning criteria, you tune the hyper parameters for that chosen criteria.
Alternatively - any indication (for example, from earlier experiments) that the best hyper parameters don’t depend on the pruning criteria?

2) RE: conclusion that gradient information is useful (for pruning criteria) in high density and not so useful in low density regimes (e.g. magnitude based pruning wins in this case) - can’t it be also a function of the batch size? I would think that for very sparse networks, increasing the batch size will improve the quality of the gradient estimates => this might result in pruning that takes gradient into account to perform better

3) Are the time complexities for all the pruning criteria the same too (I don’t think so, e.g. you need to calculate gradients for some while e.g. magnitude based pruning uses just the weights). This would be nice to include somewhere - if they all perform similar, the cheapest to calculate is preferable

Minor:
Line 21 cautions->  careful or cautious
Also I kinda find the title somewhat misleading - it suggestted to me that you will be coming up with a new criteria/method of "finding fantastic weights". Something that makes it clear that it is empirical study of pruning criteria is probably more appropriate (but I do appreciate snappy titles in general)

---

> ### Author Rebuttal · Authors · 2023-08-09
>
> **Response to Reviewer aJDG**
>
> We are grateful for the Reviewer’s feedback. We are happy to hear that the Reviewer finds that the questions we ask in our study have not been investigated before. We are also pleased to learn that our study has been recognized as rigorous and well-executed by the Reviewer. We will, of course, make our code open source (the code is also in the supplementary materials). Below, we provide answers to the raised concerns:
>
> *Limited novelty (empirical study of existing methods). Not very deep insights* - We kindly request the Reviewer's thoughtful consideration of the multifaceted nature of novelty within scientific research. A researcher's role extends beyond the creation of new methods; it encompasses an understanding of existing ones as well. Exploring established solutions serves to organize current knowledge and offers valuable insights and directions for future endeavors. Our paper exemplifies this approach by focusing on the DST framework and conducting an analysis of how diverse pruning criteria from the literature influence model performance. As aptly noted by the Reviewer, this inquiry has not been investigated before.
>
> Our work, as underscored by the Reviewer, employs a rigorous experimental setup. Notably, we demonstrate that the more intricate pruning criteria do not necessarily hold a distinct advantage over the fundamental magnitude-based approaches. Furthermore, our study indicates that in various scenarios, achieving commendable outcomes necessitates only a modest number of connectivity updates. We kindly implore the Reviewer not to confuse the simplicity of these insights with their lack of significance. Through our analysis, we cast a critical lens on the efficacy of current pruning solutions. Additionally, our results suggest that we can potentially accommodate more computationally intensive solutions for DST, given the infrequent need for updates. These contributions hold value for both present and future researchers in the field. It is noteworthy that the originality and significance of our findings have garnered praise from Reviewers xwEQ and mfmv. Considering the arguments outlined above, we sincerely invite the Reviewer to revisit our work with a broader perspective on novelty.
>
> *Mostly vision data and only 1 tabular dataset. No text data* - We add a text dataset on the task of fine-tuning ROBERTa Large (please refer to “Joint Response” for the details).
>
> **Questions:**
>
> 1. *My conceptual question is related to line 218 …* - The way the hyperparameter setup influences DST is indeed an interesting question. Our choice to use the same hyperparameter setup has two motivations. Firstly, using the same setup for all experiments allows us to assess the improvement that comes from changing only the pruning criterion in isolation from any other changes. Secondly, this choice was on par with the typical DST research in which the training hyperparameters (such as learning rate, etc.) are kept fixed, and the same as in the dense models (see for instance the MLP settings on page 4 of SET in Mocanu et al. 2018, Section 4.1 of RigL in Evci et al. 2019, Section 3.3 of SNFS in Dettmers & Zettlemoyer 2019 or page 8 of MEST in Yuan et al. 2021). We do however study a fair amount of different DST-specific hyperparameter choices. For instance, the entire section 5.3 is devoted to studying the impact of the update period hyperparameter. There we observe that the best setting of that parameter is typically common among all the different pruning criteria (Figure 4). We also study some design choices in DST such as local vs. global pruning, the impact of batch size (please see point below), and the MEST lambda hyperparameter in Appendix H, F, and A.3, respectively.
>
> 2. *RE: conclusion that gradient information is useful …* - We agree that this is an interesting question and we study it to some extent in Appendix F (we will highlight this matter more in the main text). When increasing the batch size (up to 1024 from the initial 128) we indeed observe an improvement in the performance of gradient-based criteria, especially when paired with random growth. However, the overall maximal performance of all methods decreases -- this may be the result of training with a too-large batch, as a certain level of stochasticity is considered beneficial in deep learning optimization. Note that we reuse the gradients computed in the backward pass (see also point “3” below) and hence the batch size of the training is the same as the one used to compute gradients for the pruning criterion. We have also run for the rebuttal a quick experiment where we only increase the batch size for the DST update (from 128 to 1024). The results are in Figure R5 in the Rebuttal PDF. We observe, that in general, it does not enhance performance.
>
> 3. *Are the time complexities for all the pruning criteria the same too …* - We adapt the setting from MEST (Yuan et al. 2021), in which the current gradient is used to compute the score. In consequence, the computation of such a gradient happens naturally during the backward pass and does not induce any additional cost. We will make this clear in the main text.
>
> We hope that all our answers provide the necessary clarifications to the issues raised by the Reviewer. We have also followed the Reviewer’s suggestions to include text data in the comparison. Hoping that our answers have addressed the Reviewer's concerns, we kindly ask the Reviewer to consider increasing the rating.
>
> References:
>
> Mocanu et al. "Scalable training of artificial neural networks with adaptive sparse connectivity inspired by network science." Nature communications 9.1 (2018).
> Evci et al. "Rigging the lottery: Making all tickets winners." ICML 2020.
> Yuan et al. "Mest: Accurate and fast memory-economic sparse training framework on the edge." NeurIPS (2021).
> Dettmers et al. "Sparse networks from scratch: Faster training without losing performance." arXiv:1907.04840 (2019).

---

> > ### Comment · Reviewer_aJDG · 2023-08-16
> > **Thank you**
> >
> > Thank you for your response.
> > RE: increasing the batch size experiments - I assume once you increased the batch size, you also had to retune the learning rate - or did you still kept it fixed? Fixed might be ok if you use some sort of adaptive optimizer (Adam?Adagrad), but if this is SGD, you will have to re-tune the lr and possibly the number of epochs/steps to get valid conclusions.
> > In either case, I appreciate the responses and raise my score

---

> > > ### Comment · Reviewer_aJDG · 2023-08-16
> > > **Please still consider modifying the title**
> > >
> > > I do strongly encourage you to still consider adding empirical study into the title. I do find it misleading as of now

---

> > > > ### Author Response · Authors · 2023-08-18
> > > >
> > > > We thank the Reviewer for responding to our rebuttal. We are grateful for raising the score! Regarding the increased batch size experiment: We kept the learning rate fixed and adjusted only the optimal update period (so that the total number of seen samples between updates stays the same). Indeed, as suggested by the Reviewer, the effect of the learning rate might be important and cause the generally worse performance of the increased-batch-size experiment. In response to the Reviewer's suggestion, we verify the results of this experiment with an increased learning rate up to 0.08 from 0.01 so that the learning rate to batch size ratio stays approximately the same (as suggested by the scaling laws, for instance in Smith & Quoc 2018). We present the results in the table below (mean and standard deviation out of 3 runs).
> > > >
> > > > |		 crierion         		 |   	 |   	 |   	 |   	 |  	 |  density  	 |   	 |  	 |   	 |   	 |  	 |
> > > > |:---------------------------|--------:|--------:|--------:|--------:|--------:|--------:|--------:|--------:|--------:|--------:|--------:|
> > > > |     		 |  *0.05*   |  *0.06*   |  *0.07*   |  *0.08*   |  *0.09*   |  *0.1*    |  *0.15*   |  *0.2*    |  *0.3*    |  *0.4*    |  *0.5*    |
> > > > |     		 |     |   |     |    |     |     |    |     |   | 	 | 	 |
> > > > | $\mathcal{C}_{magnitude}$         		 | 91.02 (0.05)   | **91.27** (0.19) | **91.29** (0.03) | **91.50** (0.16)    | 91.73 (0.14)  | **91.76** (0.41)  | 91.52 (0.20)  | **91.84**  (0.27) | 91.81 (0.16)  | 91.93 (0.22) | 92.04 (0.20)   |
> > > > | $\mathcal{C}_{SET}$         		 | 10.00 (0.00)     | 10.00 (0.00)  | 10.00 (0.00)  | 10.00  (0.00) | 10.00     (0.00) | 10.00 (0.00) | 91.47 (0.19) | 91.51 (0.32)  | **91.87** (0.12) | 91.79 (0.37)   | 91.97 (0.09)  |
> > > > | $\mathcal{C}_{MEST}$        		 | **91.04** (0.09) | 91.04 (0.20)| 91.18 (0.29)  | 91.23 (0.12) | **91.75** (0.24) | 91.70 (0.06) | **91.61** (0.07)   | 91.76 (0.07) | 91.79 (0.07)  | **91.99** (0.19)| 91.92 (0.06)|
> > > > | $\mathcal{C}_{RSensitivity}$| 10.00  (0.00)     | 36.00 (45.03) |  60.62 (43.87) | 10.00 (0.0) 	 | 62.12 (45.15) | 60.83 (44.06)   | 49.78  (56.25) | 89.23 (2.58) | 92.25 (0.06) | 91.74 (0.93)   | **92.20** (0.16) |
> > > > | $\mathcal{C}_{SNIP}$ 	| 90.87 (0.21) | 91.11 (0.12)   | 91.22 (0.18)   | 91.21 (0.20) | 91.42 (0.18) | 91.37 (0.27)  | 91.50 (0.17) | 91.58 (0.19) | 91.74 (0.03) | 91.73 (0.07)| 91.78 (0.34)|
> > > >
> > > > We again observe that the magnitude-based criterion performs best for low densities. The gradient-enhanced MEST also performs very well but does not hold a clear advantage over the magnitude pruning. The RSensitivity criterion is again the best for the largest studied density. We will update the Appendix to include also this experiment in section F.
> > > >
> > > > Regarding the title: We are unsure whether the changes to the title are permissible at this stage. If allowed, we will consider adjusting the title for the camera ready to "Fantastic Weights and How to Find Them: Empirical Investigation on Where to Prune in Dynamic Sparse Training."
> > > >
> > > > We again thank the Reviewer for the provided feedback and remain at the Reviewer's disposal in case of any additional concerns or questions.
> > > >
> > > > References:
> > > > Smith & Quoc, "A bayesian perspective on generalization and stochastic gradient descent." arXiv:1710.06451, ICLR 2018

---

### Official Review · Reviewer_Q7vN · 2023-07-10

**Soundness:** 2 fair
**Presentation:** 3 good
**Contribution:** 2 fair
**Rating:** 3
**Confidence:** 5

**Summary:**

This paper provides a comparison of several dynamic sparse training methods and conclude that they are perform similar unless in the ultra-sparse regime, in which case the magnitude-based pruning performs best.

**Strengths:**

The comparison is through and useful for new researchers in this field.

**Weaknesses:**

This paper only provides empirical experimental results for known dynamic sparse training. No novelty is the main issue for this paper. I think it is more proper for it to be publish in a report than in a conference proceeding.

**Questions:**

N/A

---

> ### Author Rebuttal · Authors · 2023-08-09
>
> **Response to Reviewer Q7vN**
>
> We thank the Reviewer for the feedback. We are happy to hear that the Reviewer finds our experiments thorough and sees the usefulness of our research. If we understand correctly, the Reviewer’s only concern is in the novelty of the paper.
>
> We kindly request the Reviewer to acknowledge that scientific research can encompass novelty in various forms. We strongly believe that thoroughly exploring existing solutions holds equal significance to the pursuit of new methods or architectures. Our paper presents a robust and comprehensive study that systematizes the current knowledge in the field while challenging common assumptions made in dynamic sparse training (DST), such as the perceived need for a high number of updates or the superiority of more elaborate pruning criteria. Our study establishes a strong baseline for future research endeavors. We wish to highlight that, prior to our work, there has been no comprehensive study of different pruning criteria in DST. Previous studies either examined them in isolation or intertwined them with other design choices like the growth criterion. We are the first to investigate the impact of pruning criteria on DST performance and the selection of hyperparameters. Additionally, as the Reviewer has also noted, our work provides valuable insights that can greatly benefit researchers in the field. The originality of our paper and the significance of its findings have been praised by Reviewers xwEQ and mfmv. Therefore, we sincerely ask the Reviewer to consider our work from a broader perspective of novelty and reconsider the possibility of adjusting the score accordingly. Your thoughtful consideration of these aspects will be greatly appreciated.

---

### Author Rebuttal · Authors · 2023-08-09

**JOINT RESPONSE**

We thank all the Reviewers for the time and effort taken to provide valuable insights and comments on our work. We are very glad that our research has been recognized as useful and beneficial to the community (Reviewers Q7vN, xwEQ, mfmv). Moreover, Reviewers xwEQ and mfmv appreciated the novelty and originality of our work, recognizing our comprehensive study of the DST pruning criteria. Our empirical evaluations have been praised as being thorough (Reviewers Q7vN, aJDG), well-designed, and supportive of our main claims (Reviewers mfmv, 3Q28, xwEQ).  We are also pleased to learn that the reviewers found our paper to be well-written and easy to read (Reviewers aJDG, xwEQ, mfmv, 3Q28) and did not find any major weaknesses (Reviewer xwEQ). Overall, we are delighted to receive such an encouraging response from the Reviewers regarding the aforementioned aspects of our research.

We have noted a recurrent comment from the Reviewers, suggesting an extension of our results using transformer models and/or text data. In response to this feedback, we have conducted an additional evaluation of the pruning criteria on the fine-tuning task of ROBERTa Large (Liu et al., 2019) with approximately 354 million parameters, utilizing the CommonsenseQA dataset (Talmor et al., 2018) adapted from the Sparsity May Cry (SMC) Benchmark (Liu et al., 2023). The corresponding results are presented in Table R1 within the rebuttal PDF.

We have used the exact same setup as in the SMC-Benchmark, by first performing magnitude pruning for the sparse initialization and then using DST during the fine-tuning phase.  We also use the same hyperparameters. Please note that this is a different configuration from the one used in our paper, in which we trained all the models from scratch (in spirit, the CommonsenseQA study is more similar to the EfficientNet fine-tuning experiment from Appendix G). Similar to the findings of Liu et al. (2023), we observed that in this scenario, the achievable sparsity without a significant performance decline is notably lower than in vision or tabular data scenarios. Concerning random growth, the MEST criterion appears to be the most suitable choice. For gradient growth with a density of 0.8, the RSensitivity and SET criteria initially demonstrate strong performance but are eventually surpassed by the magnitude and MEST criteria at a density of 0.7. Additionally, we noted a significantly higher variance in the outcomes across all criteria compared to vision tasks. Furthermore, we conducted a brief exploration of update period values for this problem (refer to Figure R2 in the Rebuttal PDF). Notably, we observed that overly frequent updates do not yield beneficial results. Concerning gradient growth, the most effective update period value is consistently around Δt = 500 for nearly all criteria. For random growth higher update period values generally lead to improved outcomes.

We will include these results in the paper. However, we would like to emphasize that our work already contains nine different model-dataset pairs, with model sizes varying from ~72K to ~25M parameters. We believe that our more than 7000 experiments are already a strong basis that allows us to make reasonable conclusions about the impact of the pruning criteria in DST. At the same time please note that the studied pruning criteria had been evaluated either on tabular or vision datasets in most of the works that have introduced them or used them (e.g., Mocanu et al. 2018, Yuan et al. 2021, Naji et al. 2021, Evci et al. 2020). Large language models typically have a different structure and are based on the attention mechanism, for which the applicability of sparse training is still an open area of research (Liu et al. 2023, Sun et al. 2023). This motivated our initial choice of fixing our focus on tabular and vision models.

For detailed responses to the individual concerns raised by each reviewer, please refer to our separate comments posted in response to their reviews. We have made a significant effort to address each issue and query raised by the reviewers and have included results for every suggestion made to improve our empirical evidence.

References:

Liu, Shiwei, et al. "Sparsity May Cry: Let Us Fail (Current) Sparse Neural Networks Together!." arXiv:2303.02141, ICLR (2023).
Sun, Mingjie, et al. "A Simple and Effective Pruning Approach for Large Language Models." arXiv preprint arXiv:2306.11695 (2023).
Liu, Yinhan, et al. "Roberta: A robustly optimized bert pretraining approach." arXiv preprint arXiv:1907.11692 (2019).
Talmor, Alon, et al. "Commonsenseqa: A question answering challenge targeting commonsense knowledge." arXiv preprint arXiv:1811.00937 (2018).
Mocanu et al. "Scalable training of artificial neural networks with adaptive sparse connectivity inspired by network science." Nature communications 9.1 (2018): 2383.
Evci, Utku, et al. "Rigging the lottery: Making all tickets winners." International Conference on Machine Learning. PMLR, 2020.
Yuan, Geng, et al. "Mest: Accurate and fast memory-economic sparse training framework on the edge." Advances in Neural Information Processing Systems 34 (2021): 20838-20850.
Naji, Seyed Majid, Azra Abtahi, and Farokh Marvasti. "Efficient Sparse Artificial Neural Networks." arXiv preprint arXiv:2103.07674 (2021).

---

### Decision · Program_Chairs · 2023-09-21

**Decision:**

Accept (poster)

**Comment:**

This paper provides an empirical survey of pruning and growing criteria in the dynamic sparse training framework. Notably, the simple technique of magnitude-based pruning works well. While some reviewers were concerned that this worked lacked novelty due to its focus on empirically comparing existing methods, this is not a valid concern because the paper provides a meaningful and useful contribution in the form of better understanding of the field.